# A Retrospective on the Robot Air Hockey Challenge: Benchmarking Robust, Reliable, and Safe Learning Techniques for Real-world Robotics

**Puze Liu**[1,2] **Jonas Günster**[1] **Niklas Funk**[1] **Simon Gröger**[1]
**Dong Chen**[3] **Haitham Bou-Ammar**[4] **Julius Jankowski**[5] **Ante Marić**[5]
**Sylvain Calinon**[5] **Andrej Orsula**[6] **Miguel Olivares-Mendez**[6] **Hongyi Zhou**[7]
**Rudolf Lioutikov**[7] **Gerhard Neumann**[7] **Amarildo Likmeta**[8,9]
**Amirhossein Zhalehmehrabi**[9] **Thomas Bonenfant**[9] **Marcello Restelli**[9]
**Davide Tateo**[1] **Ziyuan Liu**[3] **Jan Peters**[1,2,10]

air-hockey-challenge@robot-learning.net

## Abstract

Machine learning methods have a groundbreaking impact in many application domains, but their application on real robotic platforms is still limited. Despite the many challenges associated with combining machine learning technology with robotics, robot learning remains one of the most promising directions for enhancing the capabilities of robots. When deploying learning-based approaches on real robots, extra effort is required to address the challenges posed by various real-world factors. To investigate the key factors influencing real-world deployment and to encourage original solutions from different researchers, we organized the Robot Air Hockey Challenge at the NeurIPS 2023 conference. We selected the air hockey task as a benchmark, encompassing low-level robotics problems and high-level tactics. Different from other machine learning-centric benchmarks, participants need to tackle practical challenges in robotics, such as the sim-to-real gap, low-level control issues, safety problems, real-time requirements, and the limited availability of real-world data. Furthermore, we focus on a dynamic environment, removing the typical assumption of quasi-static motions of other real-world benchmarks. The competition's results show that solutions combining learning-based approaches with prior knowledge outperform those relying solely on data when real-world deployment is challenging. Our ablation study reveals which real-world factors may be overlooked when building a learning-based solution. The successful real-world air hockey deployment of best-performing agents sets the foundation for future competitions and follow-up research directions.

[1]Intelligent Autonomous Systems, TU Darmstadt, Germany
[2]German Research Center for AI (DFKI)
[3]Huawei German research center
[4]Huawei Noah's Ark Lab, London and University College London
[5]Idiap Research Institute, Martigny, Switzerland and EPFL, Lausanne, Switzerland
[6]Space Robotics Research Group (SpaceR), University of Luxembourg, Luxembourg
[7]Karlsruher Institut für Technologie, Karlsruhe, Germany
[8]Universita di Bologna, Bologna, Italy
[9]Politecnico di Milano, Milano, Italy
[10]Centre for Cognitive Science, Hessian.AI

38th Conference on Neural Information Processing Systems (NeurIPS 2024) Track on Datasets and Benchmarks.

# 1 Introduction

Modern machine learning techniques, particularly with the advent of Large Language Models (LLMs) and Diffusion Models (DMs), had a disrupting impact on many real-world applications, such as language processing and generation Yang et al. [2019], Brown et al. [2020], Achiam et al. [2023], board and video games Silver et al. [2016], Vinyals et al. [2019], image generation Ramesh et al. [2021], Rombach et al. [2022], Croitoru et al. [2023], and speech synthesis Prenger et al. [2019], Kong et al. [2020]. However, while researchers are trying to bring these novel approaches to the real world, purely data-driven approaches are still struggling in real-world robotics, particularly when facing dynamic tasks. Indeed, recently, different foundation models for robotics have been presented Brohan et al. [2023], Zitkovich et al. [2023], O'Neill et al. [2023], Li et al. [2024], Team et al. [2024], Kim et al. [2024]. However, to date, these models suffer from long inference times and they do not provide safety guarantees Firoozi et al. [2023], making them unsuitable for fast and dynamic real-world manipulation. Furthermore, industrial applications mostly rely on classical robotics and control techniques, relegating the data-driven methods to the area of research Peres et al. [2020], Dzedzickis et al. [2021].

Indeed, while robotics tasks are often the benchmark of choice in many areas of machine learning research, such as Computer Vision (CV) and Reinforcement Learning (RL), there is often quite a big disconnection between these benchmarks and real-world tasks: often the simulated setup is oversimplified, relieving the machine learning practitioner from the common issues arising when dealing with real robotic platforms. Thus, these benchmarks focus on small aspects of the robotics problem and do not sufficiently capture the complexity and challenges of real-world systems. This makes the deployment of machine learning solutions in real-world platforms challenging and often infeasible. To overcome these challenges and to incentivize the development of approaches that can better transfer onto real robotics systems, our Robot Air Hockey Challenge specifically focuses on a set of identified key issues preventing the deployment in dynamic real-world environments:

**Safety** Real robotics platforms live in the physical world. Therefore, the action performed by the robot must not damage the robot itself, the environment, or the people around the robot, at least at deployment time. Learning-based methods should consider safety problems and avoid dangers at any point during the execution.

**Imperfect Models** It is often challenging to obtain a good model of a real system in simulation. Particularly when dealing with black-box industrial systems that do not comprehensively describe the robot parameters and dynamics. Moreover, the real world presents many discontinuous dynamics that may heavily affect the environment's behavior. Thus, machine learning methods for real systems have to expose sufficient robustness and flexibility to deal with model mismatches.

**Limited Data** Real-world robotic data requires real-world interactions. Unlike the simulation environment, we cannot speed up the data collection process or employ massive parallelization. While large robotics datasets are already available, they are often restricted to a specific robot. It will be impractical to generalize to arbitrary platforms, as every hardware, software interface, and control system differs. As collecting real-world data is costly, difficult, and time-consuming, it is essential to develop methodologies that can adapt to the sim-to-real gap by learning from limited data.

**Reactiveness** Dynamic environments require rapid adaptation and reaction. This requirement forces the agent to be ready to react to unexpected events and to compute the control action in a real-time fashion. Often, control systems are embedded in the robot, resulting in limited computation resources, e.g., no GPU availability and a limited number of cores for the computation.

**Observation Noises and Disturbances** In real-world environments, observation noise and external disturbances will inevitably exist. If not handled properly, these noises and disturbances can seriously affect the performance of the trained model. Improving the robustness of the learning methods against observation noises and disturbances is essential for real-world deployment.

In summary, we introduced the Robot Air Hockey Challenge to provide a benchmark on a challenging, dynamic task, covering all these issues. We argue that the codified nature of the game and the limited workspace easily allow the definition of rigorous evaluation metrics, which are fundamental requirements of proper scientific evaluation. This challenge is a preliminary step towards more realistic benchmarks evaluating robust, reliable, and safe learning techniques that can be easily deployed in real-world robotic applications, going beyond the quasi-static assumption.

**Related Works**

Dynamics tasks have always been challenging benchmarks for robotics and machine learning. For example, researchers have focused on dynamics dexterity games such as ball-in-a-cup [Kawato et al., 1994, Kober and Peters, 2008], juggling [Ploeger et al., 2021, Ploeger and Peters, 2022] or diabolo [von Drigalski et al., 2021], but also on sports such as tennis [Zaidi et al., 2023], soccer [Haarnoja et al., 2024] and table tennis [Mülling et al., 2011, Büchler et al., 2022]. Robotics tasks have also been already used as benchmarks for machine learning competitions, such as the Robot open-Ended Autonomous Learning competition (REAL) [Cartoni et al., 2020], Learn to Move [Song et al., 2021], the Real Robot Challenge [Gürtler et al., 2023, Funk et al., 2021], the TOTO Benchmark [Zhou et al., 2023], the MyoSuite Challenge [Caggiano et al., 2023] or the Home Robot Challenge [Yenamandra et al., 2023]. However, most of these tasks focus on simulation or consider limited quasi-static settings, where complex real-time and safety requirements are less critical.

While we are the first to develop a structured benchmark based on robot air hockey, the task is well-known in machine learning and robotics. The first work on robot air hockey can be found in Bishop and Spong [1999], while the first learning-based approach is presented in Bentivegna et al. [2004a,b], where a humanoid robot is trained to learn air hockey skills. Due to the task complexity and versatility, it has been considered a challenging setting for evaluating robotic planning and control [Namiki et al., 2013, Igeta and Namiki, 2017, Liu et al., 2021], perception [Bishop and Spong, 1999, Tadokoro et al., 2022] and even human opponent modeling [Igeta and Namiki, 2015]. On top of that, the air hockey task has been used as a benchmark for many RL [AlAttar et al., 2019, Liu et al., 2022] and robot learning [Xie et al., 2020, Kicki et al., 2023, Liu et al., 2024] approaches. Most recently, after our NeurIPS 2023 Robot Air Hockey challenge, another benchmark on robot air hockey has been proposed by Chuck et al. [2024]. In contrast to our benchmark, it focuses more on perception, and our setting uses a much bigger playing table and requires faster and more reactive motions.

## 2 The Robot Air Hockey Challenge

The Robot Air Hockey Challenge is based on our real-world robot air hockey setup, consisting of two Kuka LBR IIWA 14 robots and an air hockey table (cf. Figure 1). The Kuka robots are equipped with a task-specific end-effector composed of an air hockey mallet connected to a rod, designed to deploy agents that do not strictly comply with the safety constraints. In addition to the real system, we also developed a MuJoCo simulation allowing for efficient testing and evaluation of the proposed control strategies. While the participants had access to an ideal version of the simulator, we evaluated the solution in a modified simulator, including various real-world factors, such as observation loss, tracking loss, model mismatch, and disturbances. Participants were allowed to

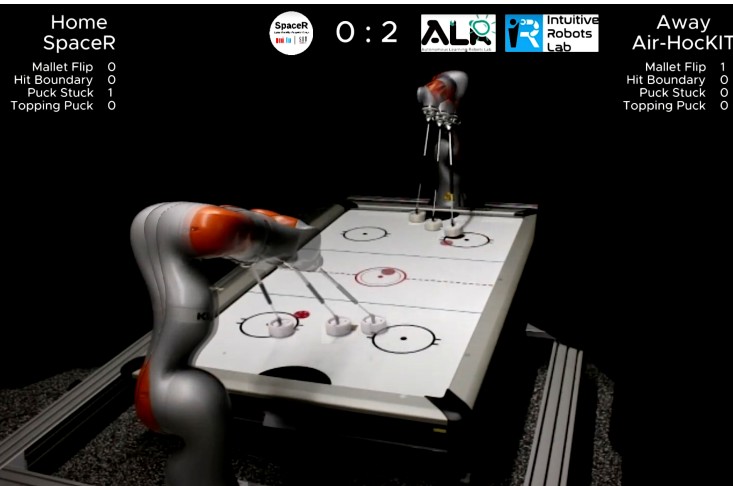

Figure 1: Game played in the real world between SpaceR and Air-HocKIT. The Air-HocKIT robot (back) hits the puck and the SpaceR robot (front) defends the attack to take control of the puck.

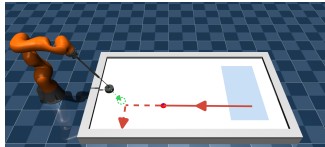 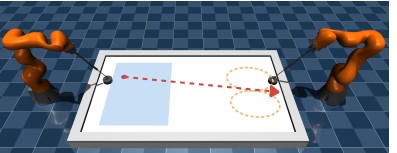 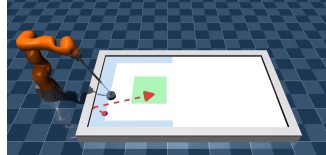

Figure 2: The three tasks in the qualifying stage of the Robot Air Hockey Challenge: from left to right "Defend", "Hit", and "Prepare" tasks. The "Defend" task requires stopping an incoming puck to get control of it. The "Hit" task consists of scoring a goal against an opponent, which moves in a fixed pattern. The "Prepare" task consists of repositioning the puck in the central area of the table without losing control of it.

evaluate their solution once per day and download the dataset obtained from the modified simulator. With this setting, we want to simulate the limited access to real-world data, forcing the participants to deal with the sim-to-real gap. On top of the sim-to-real gap, the approaches should satisfy the deployment requirements, which provide metrics to quantify whether the respective policy would be safe for real-world deployment, i.e., the policy would not harm the real setup. Details of the metrics evaluating the deployability are presented in Section 2.2.

We also provide a robotic baseline capable of playing the full match, based upon our previous work Liu et al. [2021]. The robotic baseline integrates typical approaches, such as planning and optimization. This approach performs satisfactorily in simulation but struggles to handle the sim-to-real gap. In addition, we provide a safe RL baseline Liu et al. [2022] for the low-level skills.

## 2.1 Competition Structure

The Robot Air Hockey Challenge comprises two main stages in simulation: the Qualifying and the Tournament stage. Details about the simulated environment are provided in Table 1 and Appendix A.1. In the third stage, we deployed the solutions from the top three teams in the real-world platform, validating our challenge design. The next paragraphs introduce the challenge's three stages.

**Qualifying stage** In this stage, participants need to control the robot to achieve three different sub-tasks in the single-robot environment, namely "Hit", "Defend", and "Prepare". The illustrations of the sub-tasks are shown in Figure 2.

- **Hit** The objective is to hit the puck to score a goal while the opponent moves in a predictable pattern. The puck is randomly initialized with a small initial velocity.
- **Defend** The objective is to stop the incoming puck on the agent side of the table and prevent the opponent from scoring. The puck is randomly initialized on the opponent's side of the table with a random velocity towards the agent's side, simulating an arbitrary hit from the opponent.
- **Prepare** The objective is to dribble the puck to a good hitting position. The puck is initialized close to the table boundary, where a direct hit to the goal is not feasible. The agent should maintain control of the puck, i.e., the agent must keep the puck on its half of the table.

Each task has different success metrics. For the "Hit" tasks, we count each episode as a success if the puck enters the scoring zone with a speed above the threshold. For the "Defend" task, an episode is considered successful if the final velocity of the puck (at the end of the episode) is below the threshold and does not bounce back to the opponent's side of the table. The "Prepare" task is considered successful if the final position of the puck is within a predefined area in the center of the

Table 1: Specification of the Air Hockey Environment

| Simulation Frequency | 1000 Hz |
|---|---|
| Control Frequency | 50Hz |
| Observation | Puck's X-Y Position, Yaw Angle: $[x, y, \theta]$
Puck's Velocity: $[\dot{x}, \dot{y}, \dot{\theta}]$
Joint Position / Velocity: $[q, \dot{q}]$
Opponent's Mallet Position (if applicable): $[x_o, y_o, z_o]$ |
| Control Command | Desired Joint Position / Velocity |

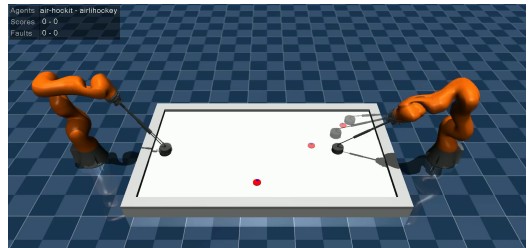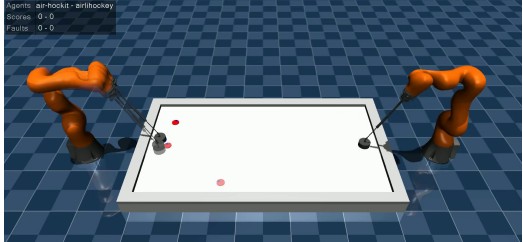

Figure 3: The tournament stage. The left figure shows the AiRLIHockey agent (right side of the table) performing a hitting motion, while the right figure shows the Air-HocKIT (left side of the table) defending the attack and taking control of the puck

agent's side of the table and the puck speed is below a given threshold. We use the mean success rate over the three sub-tasks as the overall performance metric.

**Tournament stage**    In this stage, participants trained an agent incorporating high-level skills with a low-level controller to play the full game of Air Hockey. While the qualifying stage focuses more on low-level behavior and individual primitives required to play Air Hockey, this stage requires high-level decision-making and a seamless combination of the previously developed components. The stage is composed of two rounds. In each round, we evaluate the solutions of each team against each other, such that each participant plays exactly one game against each available opponent. Before the start of each run, the authors are allowed to test their agents against the baseline agent provided by the organizers. We also run some friendly games between the currently submitted solution. Once the round started, we performed the evaluation sequentially without allowing further agent modification.

**Real-world validation stage**    Finally, we evaluated the solutions of the tournament's top three teams on our real-world air hockey platform. The deployment of the solution on the real system required another safety layer, both in terms of safety requirements and fine-tuning of the methodology and the algorithm parameters, as the simulated system still presented quite a considerable sim-to-real gap, even with domain randomization and mismatch. For this reason, for every team, the performance in the real robot system was considerably worse than in the simulated setting. Given that this stage required hand tuning and intense help and engineering from our side, the outcome of the competition relied only on the simulated tasks. Details of the safety layer can be found in Appendix A.5. Despite the sim-to-real-gap and the necessary adaptations, the agents were able to play complete games. The full videos of the matches can be found publicly at `http://air-hockey-challenge.robot-learning.net`.

## 2.2   Metrics

To evaluate the participants' solution, we focus on two important aspects: **deployability** and task **performance**. In the qualifying stage, performance is the success rate as defined above, while in the tournament stage, performance is the final game score. Based on the deployability score, teams will be categorized into three deployability levels, i.e., *deployable*, *improvable*, and *non-deployable*. Teams at the same deployability level will be ranked based on their winning score.

Table 2: Constraints for the 7DoF environment

| Constraint name | Dim. | Constraints |
|---|---|---|
| Joint position | 14 | $q_l < q_{cmd} < q_u$ |
| Joint velocity | 14 | $\dot{q}_l < \dot{q}_{cmd} < \dot{q}_u$ |
| End Effector | 5 | $l_x < x_{ee}$ 
 $l_y < y_{ee} < u_y,$ 
 $z_{ee} > z_{\text{table}} - 0.02,$ 
 $z_{ee} < z_{\text{table}} + 0.02.$ |
| Link | 2 | $z_{elbow} > 0.25$ 
 $z_{wrist} > 0.25$ |

The deployability is computed by combining various metrics. These metrics are crucial constraints that need to be respected for real-world deployment. The safety constraints are presented in Table 2. We assign a penalty point to each metric based on its level of importance. The following metrics are considered during the evaluation:

- End-Effector's Position Constraints [3 pts]: The x-y-position of the end-effector should stay inside the table's boundaries. The end-effector should remain at the table height $z_{\text{table}} = 0.1645$.
- Joint Position Limit [2 pts]: The joint position should not exceed the position limits.

- Joint Velocity Limit [1 pt]: The joint velocity should not exceed the velocity limits.
- Computation Time [0.5-2 pts]: The computation time at each step should be shorter than 0.02 s. The maximum and the average computation time (per episode) will be considered in this metric.

In the qualification stage, we will run 1000 episodes on each subtask (equivalent to 2.8 hours of actual time) to evaluate the agent. The corresponding penalty points are accumulated if any metrics are violated in an episode (up to 500 steps per episode). The deployability score (DS) is the sum of the penalty points for all episodes. During the tournament stage, a full air hockey game will be evaluated. Each game lasts 15 minutes (45,000 steps), and we treat every 500 steps as an equivalent episode. Penalty points will be refreshed at the beginning of the episode. Based on the deployability score, the ranking will be divided into three categories: *Deployable* (DS $\leq$ 500), Improvable (500 $<$ DS $\leq$ 1500), and *Non-Deployable* (DS $>$ 1500) in the qualifying stage (1000 episodes); *Deployable* (DS $\leq$ 45), *Improvable* (45 $<$ DS $\leq$ 135), and *Non-Deployable* (DS $>$ 135) in the tournament stage (equivalently 90 episodes).

## 3 Analysis

In this section, we analyze the results of the 2023 edition of the Robot Air Hockey Challenge. Despite being the first year of the competition, we received many competitive solutions and multiple solutions have been deployed in the real robot setup. In particular, it is interesting to see how the participants designed very different solutions coming from completely different perspectives. In general, many participants struggled to achieve satisfactory behaviors due to the particular challenges of the robot air hockey task. Furthermore, we identify four key outcomes of the competition:

**i.** The interplay between performance and deployability requirements makes the design of task objectives difficult. We note that, since most teams use RL in their solutions, the reward function design heavily influences the task performance.

**ii.** Penalty-based safety specification is brittle for out-of-distribution situations. Since deployability score is one important metric in the challenge, teams that adopt more engineered modules, such as inverse kinematics and trajectory interpolation, achieved better compliance with the constraints. Penalty-based training methods can satisfy safety requirements for in-distribution scenarios but may fail catastrophically for out-of-distribution situations.

**iii.** Plain RL is not sufficiently competent in handling long-horizon tasks. While it is possible to train a single agent to play a full game, the resulting approaches were not fully competitive with a human-designed policy.

**iv.** Physical inductive biases play an important role in the era of Embodiment AI. Prior knowledge such as kinematics, geometry, and physical understanding of the real world should not be ignored to obtain a safe, reliable, and robust solution deployable in the real world.

We now discuss in detail the three stages of the competition, highlighting the most relevant insights from each stage.

### 3.1 Qualifying stage

We present the aggregated results of the qualifying stage in Table 3. Additional details for the performance on each task can be found in Appendix A.3. AiRLIHockey won the qualifying stage, outperforming all other teams in all tasks. Their solution is mostly based on imitation learning, i.e., imitating a policy computed by optimal control. For the details, see Appendix A.6. The full RL solution, based on PPO, presented by the Air-HocKIT team (Appendix A.8), achieved second place both overall and in all subtasks. Both the first and second solutions achieved similar performance in almost all tasks, except the prepare task, which is quite hard to solve for RL approaches. Indeed, we observed the same behavior in previous works on similar environments [Liu et al., 2022]. The GXU-LIPE team achieved third place, however, it did not achieve third place in all tasks. Indeed, the RL3_polimi team, another RL based approach (Appdendix A.9), considerably outperformed this solution in the Defend task, while the AeroTron team was able to gain the third position on the prepare task by a short margin. It is worth noting that the gap between the top two teams and the others was quite clear. Indeed, we believe that the top teams leveraged prior knowledge and experience with robotics systems, giving them a competitive advantage against machine learning practitioners, that are not sufficiently familiar with the robotics setting. We conclude that including

robotics or physics-based inductive biases is still superior to black-box data-driven methods, and indeed most teams chose this type of solution for their final agents.

An important aspect to focus on is that many teams were able to provide satisfactory solutions in terms of performance, but struggled to comply with safety constraints. Indeed, only the ATACOM Baseline [Liu et al., 2022] can get some reasonable behavior while maintaining low safety violations. The safety issues are particularly relevant for the RL3_polimi team, which achieved the third high score but classified pretty low in the leaderboard due to safety constraints. Indeed, the black-box optimization of rule-based controllers used by the team proved to be highly sensitive to out-of-distribution evaluation. It is clear that safety issues are still a topic that is not very well explored and definitively requires more effort and investigation when working with robotic applications.

**Ablation studies**  For additional insights into the qualifying stage, we performed an ablation study on the different components causing the sim-to-sim gap between training and evaluation environments. Results of this ablation study can be found in Figure 4. We evaluated the solutions on different versions of the environment: the ideal environment without the sim-to-real gap, the original evaluation environment with all the mismatch factors activated, and four different environments that differ from the ideal one only by the addition of one of the following modifications: i. model mismatch on the robot arm, ii. observation noise, iii. disturbance on the puck dynamics (due to the airflow), iv. unreliable puck tracking (track loss). From our ablation, it is clear that model mismatch plays a marginal role in the sim-to-real gap. This is because the closed-loop controller that tracks the desired trajectory heavily mitigates the effect of the model mismatch. In addition, most of the teams applied a state observer that filters the noisy observation. Thus, the observation noise has a limited impact on the performance. Instead, tracking loss and disturbances majorly affect the outcome of the task. On average, as expected, having all mismatch factors in the environments causes the agents to perform worse. This also holds true for most solutions in most tasks. Among the tasks, the more sensitive to mismatch factors is the "Hit" one, likely due to the fast movements and sensitivity to small angular errors. Instead, a bit surprisingly, the presence of a single factor may increase the performance on a single task of the baseline. This is most notable for SpaceR and RL3_polimi solutions, e.g., under puck disturbances. We argue that the changes in the state distribution are slightly beneficial for RL solutions due to the possibility of moving the puck in areas where the policy is more competent. The same result does not apply to the Air-HocKIT team, as the success rate of this solution is already high, meaning that the policy works well in all areas of the state space. We observe a small improvement also for the AiRLIHockey and Air-HocKIT solutions when using the less relevant mismatch factors. We believe this improvement is due to the stochasticity of the evaluation and would probably disappear with more task repetitions.

### 3.2   Tournament stage

Table 3: Results from the qualifying stage

| Team Name | Hit Success | Defend Success | Prepare Success | Penalty Points | Score |
|---|---|---|---|---|---|
| Deployable Teams | | | | | |
| AiRLIHockey | 54.9% | 84.5% | 90.3% | 327.5 | 73.8 |
| Air-HocKIT | 52.2% | 79.0% | 68.6% | 341.0 | 66.2 |
| GXU-LIPE | 30.1% | 29.5% | 66.2% | 475.5 | 37.1 |
| SpaceR | 14.4% | 47.8% | 47.8% | 221.0 | 34.4 |
| Baseline | 12.8% | 25.4% | 66.3% | 352.5 | 28.5 |
| Baseline-ATACOM | 18.4% | 36.1% | 30.1% | 33.0 | 27.8 |
| AJoy | 18.4% | 36.1% | 20.0% | 108.0 | 25.8 |
| Kalash Jain | 0.0% | 19.5% | 8.3% | 0.0 | 9.5 |
| Improvable Teams | | | | | |
| RL3_polimi | 22.7% | 61.6% | 36.2% | 920.0 | 41.0 |
| AeroTron | 33.2% | 23.2% | 66.5% | 594.0 | 35.9 |
| CONFIRMTEAM | 29.3% | 23.9% | 65.3% | 629.0 | 34.4 |
| Tony | 33.2% | 23.2% | 54.3% | 718.0 | 33.4 |
| sprkrd | 0.2% | 5.5% | 0.0% | 1271.0 | 2.3 |

We present the results of the tournament stage in Table 4. Unfortunately, not all the qualified teams took part in the tournament stage, and one team only took part in one round of the competition. The winner of this stage is the AiRLIHockey team, while SpaceR scored second and Air-HocKIT scored third. The AiRLIHockey team dominated both the first and second rounds of the competition. By looking at the videos of the games, it is clear that the high

Table 4: Results from the tournament stage

| Team Name | Round 1 | Round 2 | Total Points |
|---|---|---|---|
| AiRLIHockey | 15 | 18 | 33 |
| SpaceR | 9 | 12 | 21 |
| Air-HocKIT | 5 | 15 | 20 |
| AJoy | 9 | 6 | 15 |
| RL3_polimi | 2 | 9 | 11 |
| GXU-LIPE | 2 | 1 | 3 |
| CONFIRMTEAM | 0 | 1 | 1 |

performance of this agent is mostly due to the superior hitting performance in terms of speed and accuracy. The second place was achieved by SpaceR, with a model-based RL solution based on DreamerV3. While this solution performed slightly worse than other approaches in the qualifying stage, the end-to-end training simplified the porting on the tournament environment.

Differently from SpaceR, the Air-HocKIT team relied on a high-level policy to concatenate different skills. Unfortunately, a faulty policy implementation caused this team to lose many games on the first run due to a high deployability score. A similar issue also affected the solution of the RL3_polimi team. This shows one possible limitation of modular policies, where a faulty component can cause the complete failure of the system. Indeed, in the second round, after fixing the agent in the fine-tuning phase, Air-HocKIT achieved a good score, losing only against the AiRLIHockey team. For the same reasons, also the RL3_polimi performance increased in the second round.

## 3.3   Real robot deployment

Sim-to-real transfer is a fundamental aspect of robot learning, as good performance on the real robotic system ultimately validates the presented approaches. While the real robot deployment was not part of the competition itself, we believe the sim-to-real transfer is a fundamental aspect of robot learning. In the end, we successfully deployed the top three solutions on the real robot. We recorded two full games between the agents. The deployment went on mostly smoothly, with minor issues (false score detection and mallet flipping) and both games ended up with multiple goals scored. While most of the goals originated from a mistake of the agents controlling or defending the puck, in the videos we still observe goals due to well-placed shots.

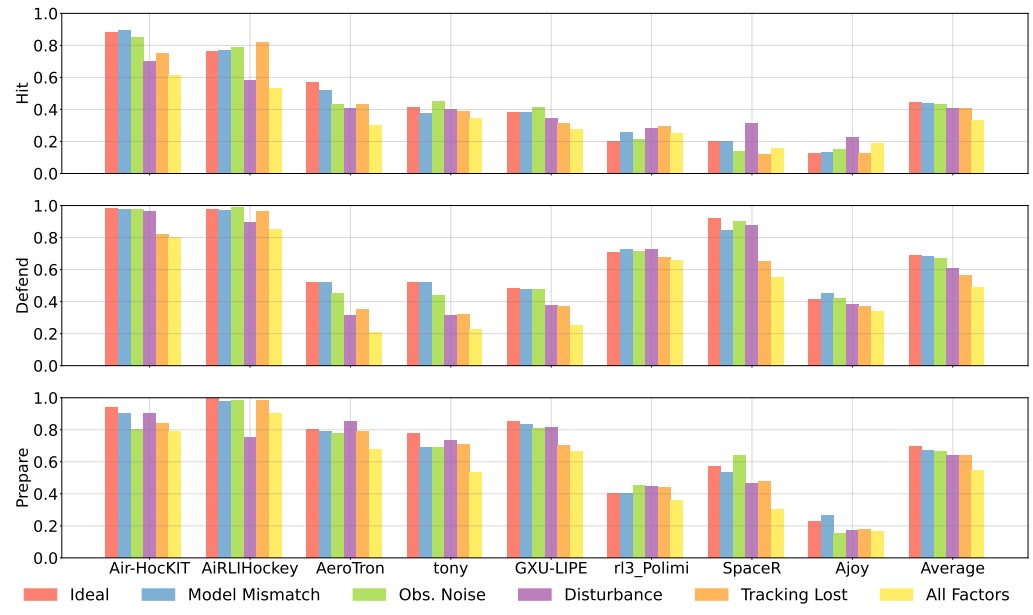

Figure 4: Ablation studies on the effect of each domain discrepancy on the submitted solutions. The graph reports the success rate for each task under different types of environment modifications

It is worth mentioning that the deployment phase was characterized by minor technical issues and required parameter tuning and adaptation for a successful deployment in the real system. Contrary to the simulation, real-world deployment is not a synchronous process, and the system is affected by delays. The delays, approximately 40ms, arise from gathering the observation, action computation, and execution. The deployment may lead to unsafe behaviors as the solutions are not trained in this environment, thus, trajectories that are reasonable in the simulation may become unsafe. Therefore, our safety layer is fundamental to avoid damaging the real platform. Details of the safety layer can be found in Appendix A.5

### 3.4 Participants solution analysis

In this section, we will briefly discuss the solutions of the AiRLIHockey, SpaceR, Air-HocKIT, and RL3_polimi teams. Detailed descriptions of these solutions can be found in the Appendix A.6, A.7, A.8, and A.9 respectively. The AiRLIHockey solution, the winner of the challenge, is heavily based on classical robotics solutions and obtains highly precise and fast-hitting motions thanks to an optimization-based solution exploiting model predictive control [Jankowski et al., 2023]. In this solution, learning is used mostly for state estimation and prediction as well as for the contact planner. The agent's high-level behavior is controlled by a state machine. The SpaceR solution, instead, is a flat policy approach based on Dreamer-V3 [Hafner et al., 2023]. This solution exploits inverse kinematics to control directly in task space, avoiding the control in joint space. This approach also avoids any type of reward shaping and exploits instead sparse reward functions and self-play to solve the task. The Air-HocKIT solution is another RL-based solution, but using a state machine to coordinate multiple low-level RL policies, trained with PPO [Schulman et al., 2017]. Differently from SpaceR, this team encodes domain knowledge in the reward function to achieve accurate, and safe low-level behaviors. The team tackles the sim-to-real gap w.r.t. the robot's motion with system identification. Finally, the RL3_polimi team also employs a hierarchical control architecture. Their state machine selects skills that are learned either with Deep RL (SAC [Haarnoja et al., 2024] combined with ATACOM [Liu et al., 2022]) or by learning rule-based policies with the PGPE algorithm [Sehnke et al., 2008], following the same scheme of Likmeta et al. [2020].

From the competition's results, it is clear that more structured solutions exploiting robotics priors perform better than unstructured ones. Experience with robotics systems is also extremely beneficial to encode prior information effectively in the reward functions or to perform parameter identification and bridge the sim-to-real gap. In general, data-driven methods are useful for learning models of the environment, for avoiding costly online optimization via behavioral cloning, and for learning dynamic behaviors. Another result is that for complex tasks, using advanced planning methods for control yields high-performance solutions that are still difficult to discover using data-driven methods, even with the help of structured reward functions. While these advanced planning methods are often computationally demanding, for limited tasks it is still possible to distill them through behavioral cloning. On one side, this shows that there are still open questions on learning for control. On the other side, this opens opportunities to use datasets generated with optimal control in combination with modern behavioral cloning and foundation models. Finally, we observe that hardcoding the high-level policy with a state machine is more beneficial than employing a flat policy. This result shows the effectiveness of specialized models against general behavioral models, at least for dynamic tasks like the one presented in this challenge.

### 3.5 Key research problems

As a final point, we exploited the very diverse backgrounds of our participants to identify the most important research questions tackled within this challenge. Indeed, the teams highlighted a wide variety of research problems such as: i. the **sim-to-real gap** and online adaptation, from the perspective of online learning or system identification; ii. the complex **contact dynamics** between the puck and the mallet; iii. the competitive **multi-agent setting** and the necessity to adapt to the opponent; iv. the design of **curriculum learning** approaches.

However, most teams highlighted two main challenges, even if analyzed from different perspectives. The first one is the **constraint satisfaction** problem. The RL3_polimi team views this problem from the point of view of exploration of RL algorithms. In particular, they view this problem as a reward-shaping problem and point out that in the future, this could be seen through the lenses of multi-objective RL. Coming from a more robotics background, the SpaceR team instead views this

problem as a low-level control problem. The Air-HocKIT team's point of view can be seen as in between the previous two, as their idea is to combine reward shaping and action space selection.

The second key research question identified is the **decision-making at different time scales** problem. Again, this issue has been seen in different ways. The AiRLIHockey team frames this problem as a control and state prediction problem, where the fast controller requires long-term prediction to act efficiently. The SpaceR and RL3_polimi teams see this issue from the perspective of Hierarchical RL.

### 3.6 Limitations

Both the challenge and the proposed analysis are affected by limitations. In particular, in our setting, we assume a good perception system: we track the puck using the Optitrack motion capture system. This assumption neglects one of the most important problems in robotics, namely perception and action-perception coupling. However, while this is a clear limitation that distances our system from generic real-world robotic tasks, we believe that the setting is already too challenging for machine learning and robotics researchers, therefore reducing the scope of the work is reasonable to obtain good solutions. In future iterations of the challenge, we may consider a more complex perception problem or add some tasks requiring direct control from camera images.

Regarding the analysis, unfortunately, our conclusions suffer from a limited sample size. This is due to the particularly challenging tasks and the lack of availability of strong baselines. In the future, we hope to lower the entrance barrier to allow for more competitors. Another issue of the analysis is that currently there is not much understanding of the importance of high-level policy. Indeed, the only insight on this aspect is that a faulty high-level policy may cause severe safety issues, resulting in game losses. We hope that, with better low-level skills baselines, we will be able to dive deeper into the high-level tactics and understand how relevant are the opponent's playing style adaptation techniques.

Finally one of the major limitations of the 2023 edition of the challenge is the limited real-world deployment due to the limited availability of the hardware, the high complexity of the task, and the complexity of the setup. This made it impossible to perform learning and data collection directly from the real world and limited our capability of testing solutions. Unfortunately, our current robotic setup neither can easily support concurrent learning of multiple solutions nor the extensive deployment of learned solutions as done in the TOTO benchmark [Zhou et al., 2023]. This is because, differently from other benchmarks, our chosen task is very complex and our system setup requires many different robotics solutions to work together. All these systems may be prone to failure and require robotic engineer supervision. For future iterations, we might consider providing real-world data and we will make the final real-world stage part of the competition. However, it is worth noting that our sim-to-sim approach closely predicts the results we can expect on real-world deployment, showing that our benchmarking approach is a viable way to evaluate real-world transfer performance.

## 4   Conclusion

In this paper, we presented a retrospective on the Robot Air Hockey Challenge. This challenge tries to bridge the gap between machine learning and robotics researchers, focusing on a domain particularly challenging for both research areas. In particular, our challenge focused on some key aspects that are often neglected by typical benchmarks used by machine learning researchers such as safety issues, dynamic environments requiring highly reactive policies and strict computational requirements, heavy sim-to-real gap, limited amount of data that can be collected from real robots, and competitive multiagent settings. While the air hockey task is particularly challenging, we were able to deploy a satisfactory number of solutions employing a variety of techniques also in the real robot. Unsurprisingly, the solution relying more on classical robotics techniques, namely model predictive control, outperformed all of the other methodologies, both in the tournament stage and in real-world deployment. Indeed, in robotics, exploiting good priors is still key to obtaining state-of-the-art performances. However, machine learning approaches are already competitive, and we are confident that a combination of data exploitation and inductive biases will allow the data-driven solution to surpass more classical baselines.

## Acknowledgments and Disclosure of Funding

The Robot Air Hockey Challenge was sponsored by the Huawei group. Organizers are partially supported by the China Scholarship Council (No. 201908080039), the EU's Horizon Europe project ARISE (Grant no.: 101135959), and the German Federal Ministry of Education and Research (BMBF) within the subproject "Modeling and exploration of the operational area, design of the AI assistance as well as legal aspects of the use of technology" of the collaborative KIARA project (grant no. 13N16274).

We acknowledge the participants who contributed to the challenge and provided insightful reports of the challenge:

- AiRLIHockey — Julius Jankowski, Ante Marić, Sylvain Calinon
- Air-HocKIT — Mustafa Enes Batur, Vincent de Bakker, Atalay Donat, Ömer Erdinç Yagmurlu, Marcus Fiedler, Zeqi Jin, Dongxu Yang, Hongyi Zhou, Xiaogang Jia, Onur Celik, Fabian Otto, Rudolf Lioutikov, Gerhard Neumann
- SpaceR — Andrej Orsula, Miguel Olivares-Mendez
- RL3_polimi — Amarildo Likmeta, Amirhossein Zhalehmehrabi, Thomas Jean Bernard Bonenfant, Alessandro Montenegro, Davide Salaorni, Marcello Restelli

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

# A  Appendix

## A.1  Simulated environment details

The details of the simulated environment are presented in Table 1 in the main paper. The observation of the environment consists of the joint position and velocity of the robot arm as well as the puck's Cartesian position and velocity, which add up to 20 dimensions in total. If the environment includes an opponent, the Cartesian position of his mallet is appended to the observation space, resulting in 23 dimensions. The simulation can be controlled with multiple different action interpolation approaches. For pure position control, we offer linear or quadratic interpolation. For position and velocity control, the commands can be interpolated by cubic, quartic, or linear interpolation. If the control command also includes acceleration quintic interpolation is used. Finally it is also possible to bypass the trajectory planar by directly supplying position, velocity and acceleration commands at 1000Hz. The robot specification, including joint limits, are presented in Table 5. The control diagram is depicted in Fig. 5. Extensive details about the environments, tasks, and interfaces together with installation instruction can be found in the online documentation at `https://air-hockey-challenges-docs.readthedocs.io/en/latest/`.

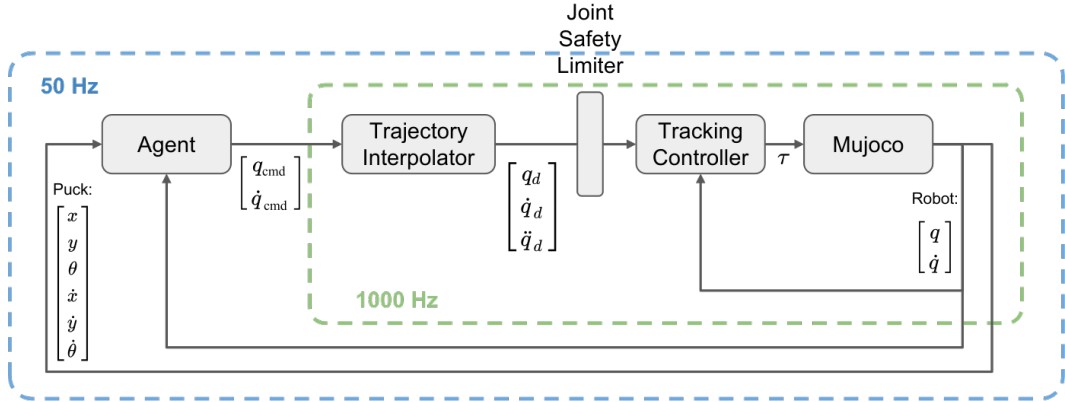

Figure 5: Control diagram of the simulated environment

Table 5: Specification of robot and environment variables

| Specification | values per joint |
|---|---|
| Joint upper limit (rad) | [ 2.967, 2.09, 2.967, 2.094, 2.967, 2.094, 3.054] |
| Joint lower limit (rad) | [-2.967, -2.094, -2.967, -2.094, -2.967, -2.094, -3.054] |
| Joint velocity limit (rad/s) | $\pm$ [ 1.483, 1.483, 1.745, 1.308, 2.268, 2.356, 2.356] |
| Initial joint position (rad) | [ 0, -0.1960, 0, -1.8436, 0, 0.9704, 0] |
| Initial joint velocity (rad/s) | [ 0, 0, 0, 0, 0, 0, 0] |

## A.2  Computation resources

The simulated evaluation is conducted on the Huawei Cloud Server using a modified environment. The cloud server is equipped with an Intel(R) Xeon(R) Gold 6278C CPU @ 2.60GHz and 48 GB RAM. To make the competition fairer and keep it accessible to a wider range of enthusiasts, the experiments are fully evaluated on the CPU. The computation time varies across the teams as different methods are used. Since the competition also evaluates the computation time per step, the experiments of the qualifying stage take around 3.5 hours, and the evaluation of the tournament takes around 40 minutes.

## A.3 Results for the Qualifying stage

Table 6: Results for the Defend task

| Team Name | Success rate | Penalty Points |
|---|---|---|
| Deployable Teams | | |
| AiRLIHockey | 84.5% | 88.5 |
| Air-HocKIT | 79.0% | 165.0 |
| RL3_polimi | 61.6% | 1.5 |
| SpaceR | 47.8% | 95.0 |
| Baseline-ATACOM | 36.1% | 33.0 |
| AJoy | 36.1% | 33.0 |
| GXU-LIPE | 29.5% | 261.0 |
| Baseline | 25.4% | 15.5 |
| CONFIRMTEAM | 23.9% | 447.5 |
| AeroTron | 23.2% | 399.5 |
| Tony | 23.2% | 420.0 |
| Kalash Jain | 19.5% | 0.0 |
| sprkrd | 5.5% | 175.0 |

Table 7: Results for the Hit task

| Team Name | Success rate | Penalty Points |
|---|---|---|
| Deployable Teams | | |
| AiRLIHockey | 54.9% | 327.5 |
| Air-HocKIT | 52.2% | 113.0 |
| GXU-LIPE | 30.1% | 475.5 |
| Baseline-ATACOM | 18.4% | 23.5 |
| AJoy | 18.4% | 23.5 |
| SpaceR | 14.4% | 35.0 |
| Baseline | 12.8% | 110.0 |
| Kalash Jain | 0.0% | 0.0 |
| Improvable Teams | | |
| AeroTron | 33.2% | 594.0 |
| Tony | 33.2% | 718.0 |
| CONFIRMTEAM | 29.4% | 629.0 |
| RL3_polimi | 22.7% | 920.0 |
| sprkrd | 0.2% | 1271.0 |

Table 8: Results for the Prepare task

| Team Name | Success rate | Penalty Points |
|---|---|---|
| Deployable Teams | | |
| AiRLIHockey | 90.3% | 132.0 |
| Air-HocKIT | 68.6% | 341.0 |
| AeroTron | 66.5% | 353.5 |
| Baseline | 66.3% | 352.5 |
| GXU-LIPE | 66.2% | 244.0 |
| CONFIRMTEAM | 65.3% | 364.0 |
| Tony | 54.3% | 443.0 |
| SpaceR | 47.8% | 221.0 |
| RL3_polimi | 36.2% | 329.0 |
| Baseline-ATACOM | 30.1% | 0.0 |
| AJoy | 20.0% | 108.0 |
| Kalash Jain | 8.3% | 0.0 |
| Improvable Teams | | |
| sprkrd | 0.0% | 1255.0 |

## A.4 Results the tournament stage

Table 9: Results of the first round of the tournament

| Match | Final Score | Goal Scored | Penalty Points |
|---|---|---|---|
| **AiRLIHockey** × GXU-LIPE | 10-0 | 10-0 | 62.0 / 384.0 |
| **AiRLIHockey** × SpaceR | 13-2 | 13-2 | 60.0 / 151.0 |
| **AiRLIHockey** × RL3_polimi | 8-0 | 7-0 | 59.0 / 295.5 |
| **AiRLIHockey** × AJoy | 37-0 | 33-0 | 54.0 / 41.0 |
| **AiRLIHockey** × Air-HocKIT | 14-3 | 14-3 | 68.0 / 333.5 |
| RL3_polimi × **AJoy** | 18-2 | 8-0 | 217.5 / 36.0 |
| RL3_polimi × Air-HocKIT | 3-12 | 3-12 | 227.0 / 329.5 |
| RL3_polimi × **SpaceR** | 0-5 | 0-1 | 321.0 / 110.0 |
| RL3_polimi × GXU-LIPE | 1-5 | 1-4 | 196.0 / 316.5 |
| Air-HocKIT × GXU-LIPE | 8-2 | 8-2 | 358.0 / 370.5 |
| Air-HocKIT × **AJoy** | 57-2 | 52-2 | 238.5 / 40.0 |
| **Air-HocKIT** × SpaceR | 24-8 | 24-8 | 93.5 / 121.0 |
| **SpaceR** × AJoy | 17-0 | 2-0 | 25.0 / 35.0 |
| **SpaceR** × GXU-LIPE | 3-1 | 3-1 | 79.0 / 387.0 |
| **AJoy** × GXU-LIPE | 4-30 | 4-23 | 45.0 / 294.0 |

Table 10: Results of the second round of the tournament

| Match | Final Score | Goal Scored | Penalty Points |
|---|---|---|---|
| AJoy × **AiRLIHockey** | 1-64 | 1-61 | 31.0 / 77.0 |
| RL3_polimi × **AiRLIHockey** | 0-39 | 0-39 | 73.0 / 66.0 |
| Air-HocKIT × **AiRLIHockey** | 0-4 | 0-4 | 3.0 / 77.0 |
| GXU-LIPE × **AiRLIHockey** | 1-9 | 1-9 | 363.0 / 69.0 |
| CONFIRMTEAM × **AiRLIHockey** | 1-57 | 1-57 | 387.5 / 74.0 |
| SpaceR × **AiRLIHockey** | 0-23 | 0-23 | 94.0 / 78.0 |
| **Air-HocKIT** × RL3_polimi | 4-0 | 4-0 | 15.0 / 65.0 |
| GXU-LIPE × **RL3_polimi** | 21-0 | 20-0 | 321.5 / 73.0 |
| AJoy × **RL3_polimi** | 2-23 | 2-12 | 28.0 / 44.0 |
| CONFIRMTEAM × **RL3_polimi** | 2-1 | 2-1 | 368.0 / 95.0 |
| **SpaceR** × RL3_polimi | 9-0 | 8-0 | 20.0 / 62.0 |
| AJoy × **Air-HocKIT** | 1-94 | 0-89 | 33.0 / 12.0 |
| GXU-LIPE × **Air-HocKIT** | 1-8 | 1-8 | 343.5 / 13.0 |
| SpaceR × **Air-HocKIT** | 4-23 | 3-23 | 76.0 / 17.0 |
| CONFIRMTEAM × **Air-HocKIT** | 3-9 | 3-9 | 376.0 / 12.0 |
| GXU-LIPE × **SpaceR** | 7-1 | 7-1 | 348.5 / 85.0 |
| CONFIRMTEAM × **SpaceR** | 6-2 | 6-1 | 371.0 / 91.0 |
| AJoy × **SpaceR** | 0-9 | 0-2 | 34.5 / 7.0 |
| CONFIRMTEAM × **AJoy** | 49-1 | 42-1 | 299.5 / 36.0 |
| GXU-LIPE × **AJoy** | 41-0 | 33-0 | 277.0 / 37.0 |
| CONFIRMTEAM × GXU-LIPE | 5-0 | 5-0 | 384.0 / 346.5 |

Table 11: Aggregate metrics for the tournament

| Team Name | Wins | Loses | Draws | Goals Scored | Goals Received | Penalty Points |
|---|---|---|---|---|---|---|
| AiRLIHockey | 11 | 0 | 0 | 270 | 8 | 744.0 |
| SpaceR | 7 | 4 | 0 | 31 | 97 | 859.0 |
| Air-HocKIT | 6 | 3 | 2 | 232 | 40 | 1425.0 |
| AJoy | 5 | 6 | 0 | 10 | 357 | 396.5 |
| RL3_polimi | 3 | 6 | 2 | 25 | 99 | 1669.0 |
| GXU-LIPE | 0 | 8 | 3 | 92 | 49 | 3752.0 |
| CONFIRMTEAM | 0 | 5 | 1 | 59 | 69 | 2186.0 |

## A.5 Real-world Deployment

In this section, we highlight the changes we performed to adjust the solutions for real-world deployment.

To increase safety we designed a compliant end effector, composed of a metal rod connected to a gas spring. This rod is connected to a (non-actuated) universal joint, that allows for a larger workspace. Finally, the mallet itself is compliant, allowing for a couple of centimeters of additional compression, thanks to a foam core.

As already discussed, the observations were gathered from the robot's motor encoders and an Optitrack motion capture system. We use the information about the mallet position and orientation from the forward kinematics of the joint position and the Optitrack motion capture system to check if it is flipped, pressed too far into the table, or hitting the table boundaries. Indeed, the presence of the universal joint may cause unwanted flipping of the mallet.

Before executing an action it is checked for safety to prevent the robot from hitting the table surface or boundaries. We also check if the velocity and acceleration required to satisfy the requested setpoint command are achievable by the robot. To prevent the robot from pushing too hard into the table or lifting the end-effector and, consequently, flipping the mallet, we use an Inverse Kinematics approach to adjust the action, ensuring that the end-effector will keep the table height. This controller can correct small errors and lead to a much smoother behavior of the robot but cannot correct larger errors or large vertical velocities, mainly due to the robot's physical limitations. When a mallet flip is detected, the safety layer interferes, lifts the mallet, moves the robot back to its initial configuration, and restarts that robot's episode, and then the agent can continue performing the current task. In the other safety-critical cases, which occur less frequently, the game is paused and restarted after the robots have been moved to their initial position.

The control system generates a trajectory for the next 100ms using the point action provided by the agent. The trajectory starts at the current state of the robot at t=0, the drawn action is set at t=20ms and it is extended for the next 80ms with a constant velocity model. Normally, a new trajectory should arrive after 20ms, and replace the currently executed one, however, this solution prevents jerky motion in case of action delays.

The torque output is computed by an Active Disturbance Rejection Controller (ADRC) with a feed-forward term. We interpolate the trajectory linearly in position, velocity, and acceleration. While the resulting trajectory is not physically consistent and impossible to reproduce, we experimentally found that, for high-frequency trajectories, this behavior reduces oscillating torques compared to quintic polynomial interpolation, commonly used in standard robotics control toolkits.

## A.6 Details of the AiRLIHockey solution

The *AiRLIHockey* agent addresses the uncertainty and long-horizon nature of air hockey by relying on an optimal control formulation of primitive skills such as *shooting*, *defense*, and *preparation* to generate reactive behavior. While defense and preparation can be addressed through simple heuristics, shooting requires reasoning over longer prediction horizons. Thus, to operate at a 50 Hz replanning rate, offline precomputation is leveraged with an energy-based model trained to quickly recover shooting actions. An additional key to the performance of the solution lies in fast optimization of subsequent mallet trajectories through zero-order methods combined with low-dimensional trajectory

representations [Jankowski et al., 2023]. These trajectories further maximize the shooting velocity while ensuring task space and joint constraints. Fig. 6 illustrates the framework.

### A.6.1 State Estimation & Prediction

Puck dynamics are modelled as piecewise (locally) linear with three different modes: **(a)** *The puck freely sliding on the table*, **(b)** *The puck is in contact with the wall*, and **(c)** *The puck is in contact with the mallet*. Data collected in simulation is used to learn linear parameters $\boldsymbol{A}_i, \boldsymbol{B}_i$, and an individual covariance matrix $\boldsymbol{\Sigma}_i$ representing the process noise for each mode. This provides a probability distribution over puck trajectories

$$\Pr_i(\boldsymbol{s}^p_{k+1}|\boldsymbol{s}^p_k, \boldsymbol{s}^m_k) = \mathcal{N}\Big(\boldsymbol{A}_i\boldsymbol{s}^p_k + \boldsymbol{B}_i\boldsymbol{s}^m_k, \boldsymbol{\Sigma}_i\Big), \tag{1}$$

with puck state $\boldsymbol{s}^p_k = (\boldsymbol{x}^p_k, \dot{\boldsymbol{x}}^p_k)$ and mallet state $\boldsymbol{s}^m_k = (\boldsymbol{x}^m_k, \dot{\boldsymbol{x}}^m_k)$. While each of the modes is a Gaussian distribution, marginalizing over the mallet state to propagate the puck state is not possible due to the discontinuity stemming from contacts. Therefore, an extended Kalman filter is utilized to estimate puck states across different timesteps for any given mallet state.

### A.6.2 Contact Planning

Depending on the desired behavior, the robot is required to generate a *shooting*, *defense*, or *preparation* contact plan and corresponding trajectory. These subproblems are addressed separately.

**Shooting.** Due to process and observation noise, contact planning for the shooting behavior is posed as a stochastic optimal control problem, and simplified through the following assumptions:

1. The initial timestep of the planning horizon $k = 0$ corresponds to the time of contact between the puck and mallet.

2. The cost metric is evaluated only for the final timestep of the horizon $k = K$ where the puck is at the goal line.

3. Time of contact is preset and is not a decision variable.

4. Mallet velocity at contact time is always maximized in the shooting direction while respecting dynamic constraints.

Keeping these assumptions in mind, a stochastic optimal control problem is formulated with the aim to minimize a function of the probability distribution of the puck state. To compute the cost of a candidate contact state, the probability distribution over puck trajectories is approximated as described in Section A.6.1. The cost is defined as a weighted sum of the probability of scoring a goal and expected puck velocity at the goal line, with an additional penalty term on trajectories with a low probability of entering the goal. Different trade-offs between puck velocity and scoring probability can be achieved by tuning the cost coefficients. The input space is further reduced to a single dimension by parametrizing the mallet position as a shooting angle relative to the puck. To solve the posed optimization problem at the same control rate of 50 Hz as the mid-level controller, the heavy computational burden is transferred to an offline phase in which optimal plans for a variety

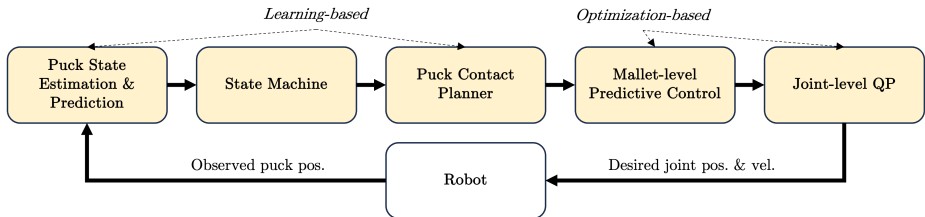

Figure 6: Overview of the *AiRLIHockey* agent. Starting from noisy observations, the puck state is estimated subject to learned model parameters. Different behaviors are triggered by a heuristic state machine based on estimated puck states. Contact and trajectory planners subsequently generate a desired action sequence that is tracked on the joint level with constrained quadratic programming.

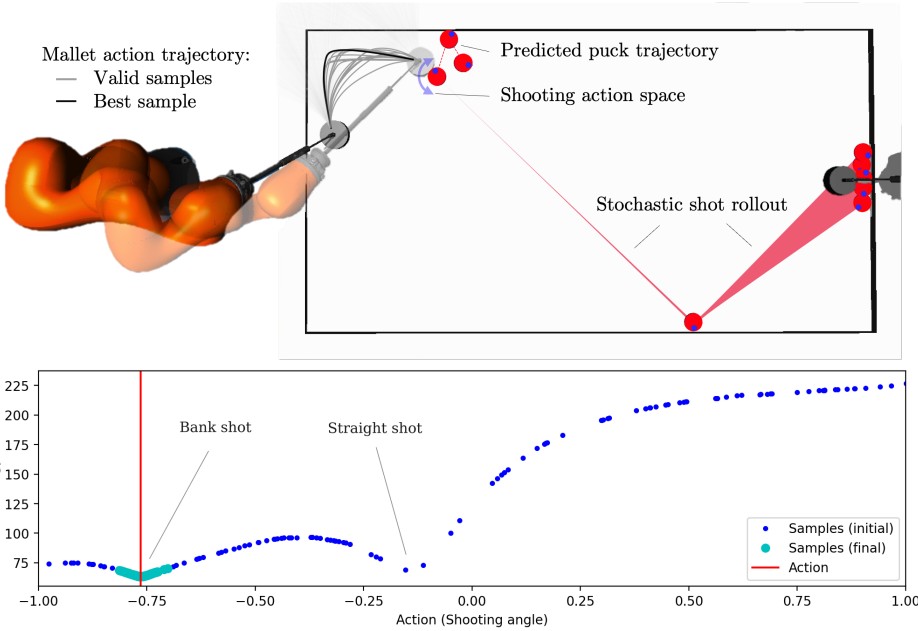

Figure 7: *Above:* overview of the interplay between the puck and mallet in scoring a goal. Given the mallet position and the estimated puck position, our framework generates a motion plan for the mallet such that the probability of scoring is maximized. *Below:* action space sampling for the displayed scenario. Dark blue points show samples at the initial timestep. Iterative recentering and variance reductions lead to the final samples shown in cyan. The red vertical line denotes the selected action.

of scenarios are collected and used to train an energy-based behavior cloning model [Florence et al., 2022] to rapidly recover optimal shooting angles $\hat{a}$ online:

$$\hat{a} = \underset{a \in \mathcal{A}}{\operatorname{argmin}} \, E_\theta(\boldsymbol{s}_0^p, a), \tag{2}$$

where $\boldsymbol{s}_0^p$ represents an initial puck state. At runtime, this is achieved by iteratively sampling a number of candidate actions at each timestep with recentering and reductions on the sampling variance. Finally, the state-action pair with the lowest energy $E_\theta$ is selected, as shown in Figure 7. Depending on the puck state, it can be observed that the optimal shooting angle produces bank shots as a consequence of rewarding high puck velocities at the goal line. This is due to the kinematics of the robot arm that is able to generate higher mallet velocities for bank shots while satisfying joint velocity limits.

**Defense & Preparation.** To generate a mallet state for deflecting the puck away from the goal, a sampling-based optimization technique is applied online. The objective of the optimization is to achieve a desired vertical puck velocity after contact (e.g. close to zero). The same optimization technique is used to prepare for a shot, with the goal of moving the puck toward the horizontal center by reflecting it against the wall. The target puck state is computed heuristically based on the puck position at the time of the contact.

### A.6.3   Mallet Trajectory Planning & Control

The mid-level control layer is responsible for executing planned contacts without colliding with the table walls. The desired mallet state at a given point in time therefore serves as a constraint for the trajectory controller. For lower computational burden, the dimensionality of the decision variable is reduced by using a basis function trajectory parameterization that ensures most of the constraints [Jankowski et al., 2023]. The utilized parameterization also minimizes an acceleration functional to generate smooth trajectories. Furthermore, since the basis functions are computed offline, a significant part of the computational burden is again transferred from online control loops. The final mallet velocity is used as a decision variable and the error w.r.t. to the desired velocity as the new objective. The optimal trajectory is generated stochastically by sampling a number of

Table 12: Observation space of the SpaceR agent

| State | Dimension |
|---|---|
| Participant — Joint positions | 7 $(DoF)$ |
| Participant — Mallet position | 2 $(x, y)$ |
| Opponent  — Mallet position | 2 $(x, y)$ |
| Puck position | 2 $(x, y)$ |
| Puck orientation | 2 $(sin, cos)$ |
| Fault timer | 1 $(t)$ |

candidate mallet velocities and evaluating the corresponding rollouts (see Fig. 7). Finally, the first mallet action leading to the lowest-cost trajectory is applied as a control reference.

## A.7 Details of the SpaceR solution

The SpaceR agent revolves around leveraging model-based RL to acquire a policy capable of playing the full game of robot air hockey. DreamerV3 [Hafner et al., 2023] is employed as the underlying algorithm to concurrently learn a world model while optimizing actor and critic networks from abstract trajectories generated by the model. The agent is guided towards optimal behaviour purely through sparse rewards corresponding to relevant score-affecting events that are not biased by a potentially complex reward shaping. In order to acquire a policy capable of competing against various approaches, self-play is employed to support the agent in discovering more robust strategies by exploiting and correcting any weaknesses in its prior behaviour.

### A.7.1 Observation Space

The observation space of the agent is summarized in Table 12 and captures all low-dimensional environment states that are available to the participants, including the state of the controlled robot, opponent robot, and puck. Additionally, the duration of the puck spent on either side of the table is encoded in the observation space to provide the agent with time awareness and avoid faults. All positions are normalized to the range $[-1, 1]$ based either on the joint limits or dimensions of the table. The orientation of the puck is encoded as sine and cosine of the angle to preserve its continuity, while the duration of the puck spent on either side of the table is normalized based on the time limit until a fault would be accumulated with the sign corresponding to the side of the table.

### A.7.2 Action Space

The agent interacts with the environment through continuous high-level actions that express the absolute target position of the mallet as a 2D vector. The target position is normalized to the range $[-1, 1]$ within the intersection of the table and the reachable workspace of the robot. The high-level actions are then mapped into lower-level joint space commands using the Inverse Jacobian method. Initially, the relative displacement of the mallet $(\Delta x, \Delta y, \Delta z)$ is determined from its current position to the target, with the position along the Z-axis constrained to the relative height of the table. The target joint displacements are calculated as $(\Delta\theta_1, \Delta\theta_2, ..., \Delta\theta_7) = J^+ \cdot (\Delta x, \Delta y, \Delta z)$ using the pseudo-inverse of the Jacobian matrix $J^+$ while emphasizing the translation along the Z-axis to maintain consistent contact of the mallet with the table. The joint displacements are clipped based on the joint velocity limits to ensure that the motion is dynamically feasible. Finally, the resulting joint positions and velocities are derived through linear interpolation over the duration of the control cycle.

### A.7.3 Reward Function

The reward function that guides the behaviour of the SpaceR agent is designed to provide sparse signals corresponding to the primary score-affecting events of the game, namely scoring a goal, receiving a goal, and causing a fault. Although a positive reward could also be attributed when the opponent receives a fault, this event is not considered due to the potential ambiguity in credit assignment. The summary in Table 13 illustrates three distinct configurations of the reward function that were adopted to reinforce agents towards different strategies in the form of *balanced*, *aggressive*, and *defensive* playstyles. The *balanced* strategy is designed to encourage the agent to play a rounded game by scoring goals while defending their side of the table. By providing a higher reward for

scoring a goal, the *aggressive* strategy incentivizes the agent to take more risks for an opportunity to score. In contrast, the *defensive* strategy does not provide any reward for scoring a goal to encourage the focus on preventing the opponent from scoring.

Table 13: Components of the SpaceR reward function that reinforce different strategies

| Strategy | Score a goal | Receive a goal | Cause a fault |
|---|---|---|---|
| Balanced (default) | $+\frac{2}{3}$ | $-1$ | $-\frac{1}{3}$ |
| Aggressive | $+1$ | $-1$ | $-\frac{1}{3}$ |
| Defensive | $0$ | $-1$ | $-\frac{1}{3}$ |

### A.7.4 Self-Play

To overcome the limitations of training solely against the baseline, the SpaceR agent employs a form of fictitious self-play to iteratively discover more robust strategies. Each agent is trained against a pool of opponents with frozen weights, with a new opponent uniformly sampled at the beginning of each episode. The pool is gradually expanded with a new model every 1000 episodes up to a total of 25 opponents. Additionally, the strategies described in the previous section are incorporated into self-play using a two-step procedure. First, three agents following the distinct strategies are trained using self-play against an opponent pool initialized with the baseline agent. Once these agents exhibit stable behaviour, their training is terminated while keeping a history of their checkpoints. Subsequently, a new agent is trained from scratch following the *balanced* strategy with the opponent pool pre-filled with the checkpoints of the previously trained agents. In this way, the new agent is immediately exposed to a diverse pool of advanced opponents with expertise in different aspects of the game, enhancing the discovery of more robust strategies.

### A.7.5 Multi-Strategy Ensemble

Recognizing the distinct strengths and weaknesses exhibited by different strategies, the SpaceR agent forms an ensemble that dynamically switches between them based on the estimated score of the match. By default, the *balanced* strategy is used for the majority of the game. If the opponent is in the lead, the ensemble switches to the *aggressive* strategy to increase the likelihood of scoring a goal through a more risky playstyle. Conversely, if the opponent is trailing by a significant margin, the *defensive* strategy is activated to provide a safer alternative in preventing the opponent from scoring a goal. With this design, the intended effect of the ensemble is to maximize the chances of winning by adapting the playstyle of the agent to the current state of the game.

### A.8 Details of the Air-HocKIT solution

The Air-HocKIT agent was constructed by integrating five independent PPO Schulman et al. [2017] models, each specialized in handling specific scenarios: hit, fast defend, slow defend, close prepare, and far prepare. Governed by a hand-crafted state machine, the selection of an agent at each time step was dictated by the current observation. A Jacobian-based reset agent was implemented to recover the robot to its original joint configuration before switching to a new agent. Each sub-agent was trained separately with a customized reward function. The main ideas behind each reward function are outlined below.

### A.8.1 Rewards Design for Agents

**Hit Agent**. The reward function for the hitting model consists of three separate stages. Before the robot hits the puck, moving the end effector towards the puck's position is rewarded. Following contact between the end effector and the puck, a larger reward, linearly scaled with the puck's x-velocity, is provided. Lastly, a substantial reward is granted for scoring a goal, multiplied by the puck's velocity and an additional base reward. The scoring reward is several magnitudes larger than all the previous step rewards, which becomes the main objective of the model after sufficient training

time. The reward for the hitting agent is defined as

$$r_{hit} = \begin{cases} \max(0, \frac{p_p - p_{ee}}{|p_p - p_{ee}|} \cdot v_{ee}) & \text{if } |v_p| < 0.25 \text{ and } p_{p,x} < 0, \\ 10 \cdot |v_p| & \text{if hit}, \\ 2000 + 5000 \cdot |v_p| & \text{if scored}, \end{cases} \quad (3)$$

where $p_p$ and $v_p$ represent the puck position and velocity, respectively, the $p_{ee}$ and $v_{ee}$ states the end-effector position and velocity.

**Defend Agents**. The two defend agents are separated by the incoming puck's velocity. The reward calculation for the slow defend strategy involves two parts. In the first part of the reward in (4), a modest positive reward is granted when the end-effector contacts the puck for the first time. The value consists of a constant term and an exponential bonus term to encourage smaller puck velocity after contact. The reward for the slow defend agent is defined as

$$r_{\text{defend\_slow}} = \begin{cases} 30 + 100^{1 - 0.25|v_p|} & \text{if } v_p > -0.2 \text{ and ee touches puck for the first time}, \\ \cdots + 70 & \text{if } |v_p| < 0.1 \text{ and } -0.7 < p_{p,x} < -0.2 \text{ and } t = T, \\ 0.01 & \text{otherwise}. \end{cases} \quad (4)$$

A binary reward was used for the fast-defend agent. The agent receives a negative reward with a value of $-100$ for being scored by the opponent. The reward remains 0 for all the other cases.

**Prepare Agents**. The two preparation agents are distinguished by the puck's proximity to our goal along the x-direction. For the close preparation, the agent is rewarded for moving the puck to a pre-defined target position, e.g., $(-0.5, 0)$ in our case, plus a small reward for moving the end-effector towards the puck. Additionally, a large reward of 2000 is granted when the success criterion is fulfilled. The reward for close preparation is defined as

$$r_{\text{proximity}} = \begin{cases} \max(0, \frac{p_p - p_{ee}}{|p_p - p_{ee}|} \cdot v_{ee}) & \text{if } |v_p| < 0.25 \text{ and } p_{p,x} < 0, \\ 0 & \text{otherwise}, \end{cases} \quad (5)$$

$$r_{\text{bonus}} = \begin{cases} 2000 & \text{if } |v_p| < 0.5, -0.65 < p_{p,x} < -0.35 \text{ and } -0.4 < p_{p,y} < 0.4, \\ 0 & \text{otherwise}, \end{cases} \quad (6)$$

$$r_{\text{close\_prepare}} = r_{\text{proximity}} + r_{\text{bonus}} + 10 \max(0, \min(0.5, \frac{(-0.5, 0)^T - p_p}{|(-0.5, 0)^T - p_p|} \cdot v_p)). \quad (7)$$

In contrast, the far prepare agent aims solely to return the puck to the opponent while avoiding faults. This approach stems from our observation that the PPO algorithm encounters challenges when exploring action sequences requiring the end effector to first maneuver to the puck's back-side for preparation. As a result, the reward function (8) is identical to the hit reward with the caveat that the large reward is bound to getting the puck far enough into opponent territory without any puck velocity bonus, upon which the episode ends. The reward function for far preparation is given as

$$r_{\text{far\_prepare}} = \begin{cases} \max(0, \frac{p_p - p_{ee}}{|p_p - p_{ee}|} \cdot v_{ee}) & \text{if } |v_p| < 0.25 \text{ and } p_{p,x} < 0, \\ 3000 & \text{if } p_{p,x} > 0.2, \\ 10 \cdot |v_p| & \text{otherwise}. \end{cases} \quad (8)$$

### A.8.2 State Machine

An overview of the hand-crafted state machine is provided in Figure 8.

### A.8.3 Mitigating Sim2Real Gap

**Observation Noise.** The observation noise during evaluation consists of additive noise on the puck positions and velocities and loss of puck tracking. The latter means that the observed puck position stays constant and hence, the observed puck velocity is zero. This is a major change for the agent as this case does not appear during training. In the Air-HocKIT solution, the additive noise and loss of tracking are modelled in the training environment. The additive noise was modeled with a Gaussian distribution that is estimated with maximum likelihood from observed and smoothed puck trajectories. Additionally, a Kalman filter is applied to the observations [Kalman, 1960] and provide the estimated

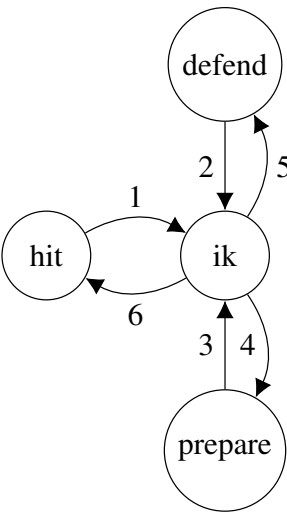

Figure 8: **State Machine of the composite agent.** The figure visualizes the different states of the the composite agent. In states *hit, defend, prepare* the hitting, defending and preparation agent is activated respectively. In the *ik* state the agent is supposed to go back to the initial joint position from which it can transition to the different states. The conditions for transitioning are shown in Table 14.

Table 14: **State transition conditions for the state machine in Fig. 8.**

| Condition | Expression |
|---|---|
| 1 | $\Delta p_{p,x} = p_{p,x} - 1.51$ 
 $(\Delta p_{p,x} > -0.2) \vee (\Delta p_{p,x} + \frac{1}{2}v_{p,x} > -0.2) \vee (p_{p,x} \leq p_{ee,x}) \vee (|p_{p,y}| > m) \vee |p_{p,y} + 0.75 v_{p,y}| > m$ |
| 2 | $(v_{p,x} > -0.2) \vee (p_{p,x} < p_{ee,x})$ |
| 3 | $|p_{p,y}| < 0.41 \vee p_{p,x} > -0.2$ |
| 4 | $[(\Delta p_{p,x} < -0.2) \wedge (\max(|v_{p,x}|, |v_{p,y}|) < 0.05)] \wedge$ 
 $[(\Delta p_{p,x} \leq -0.8) \vee |p_{p,y}| > m]$ |
| 5 | $[((\Delta p_{p,x} < 0.3) \wedge (v_{p,x} < -0.5)) \vee (v_{p,x} < -1.5)] \wedge (p_{ee,x} < p_{p,x})$ |
| 6 | $[(\Delta p_{p,x} < -0.2) \wedge (\Delta p_{p,x} + v_{p,x} < -0.2)] \wedge (v_{p,x} < 0.5) \wedge (|v_{p,y}| < 0.5)$ 
 $\wedge \neg [(|p_{p,y}| > m) \vee (|p_{p,y} + 0.75 v_{p,y}| > m)] \wedge (\Delta p_{p,x} + 0.75 v_{p,x} > -0.8)$ |

mean position as observation to the agent. The observed puck position and the estimated positions are provided in case tracking of the puck is lost. The puck's velocity is estimated by using the finite differences of the estimated puck position.

**Model dynamics and controller characteristics.** For the same initial conditions, the observed robot movements during evaluation differ from the movement during the training environment. In recorded data, this difference can be up to 1cm, which is critical for tight constraints such as staying within 2cm of the table surface. This mismatch can be explained by differences in the model and controller parameters, including masses, friction coefficients, damping coefficients, and proportional and differential gains of the controller. Air-HocKIT solution includes estimating the correct values of these entities by framing it as a black-box optimization problem. The objective is to minimize the differences between observed and simulated joint movements. This problem was solved using the Covariance Matrix Adaptation Evolution Strategy algorithm [Hansen, 2023].

**Hyperparameters**. Key hyperparameters for PPO agents are listed below. All other parameters remain consistent with those in stable-baselines3. [Raffin et al., 2021]

Table 15: List of hyperparameters.

| Hyperparameter | Value |
|---|---|
| Number of environments | 40 |
| Number of steps | 512 |
| Batch size | 512 |
| Learning rate | $5 \cdot 10^{-5}$ |
| Gamma [Defend] | 1 |
| Gamma [Hit, Prepare] | 0.99 |
| Number of epochs | 10 |
| Network architecture | [64, 64] |

## A.9 Details of the RL3_polimi solution

The *RL3_polimi* team addresses the Air Hockey problem by employing Deep RL techniques and RL Optimized Rule-Based Controllers. Indeed, they propose to train several low-level agents to perform some basic tasks in the game, and combine them via a rule-based controller to play the whole game. In what follows, we present how *RL3_polimi* models the Air Hockey environment as an MDP, the employed Deep RL techniques, the developed Rule-Based controllers optimized via RL, and finally , how they treat the noise in the evaluation environment.

### A.9.1 AirHockey as an MDP

In this section, we present how *RL3_polimi* models the Air Hockey environment as an MDP by defining the state-action space and reward functions employed.

*State Space:* The *RL3_polimi* team slightly modifies the observation provided by the original environment. Indeed, they decided to discard the rotation-axis elements of the original observation, add the x and y component of the end effector computed with forward kinematics and add a flag indicating whether a collision happened between the end-effector and the puck. The resulting state space is then scaled between -1 and 1.

$$s = [\mathbf{p}_{\text{puck}}\ \dot{\mathbf{p}}_{\text{puck}}\ \mathbf{q}\ \dot{\mathbf{q}}\ \mathbf{p}_{\text{ee}}\ \dot{\mathbf{p}}_{\text{ee}}\ \text{collision\_flag}]$$

To smooth out the noise in the puck position and velocity observation, *RL3_polimi* employed a simplified Kalman Filter [Kalman, 1960]. In order to deal with noise in the joint positions and velocities, they directly use the agents actions in the previous step and ensure the requested robot poses are valid.

*Action Space: RL3_polimi* employs a high-level control setting where the agent controls either acceleration of the joints or directly the end-effector position which are then converted to joint position and velocities depending on the low-level task.

*Reward function:* The designed *reward* is task-dependent, given that *RL3_polimi* trains low-level policies for each low level task. Moreover, they shape the reward function instead of only providing a sparse reward for the completion of these tasks. Since this reward shaping greatly affects the performance of the learning algorithm, they employ different reward functions for the DeepRL and Rule-based agents. Nevertheless, in each case, the reward function follows the following general structure:

$$r(s, a) = r_t(s, a) + r_c(s, a),$$

where $r_t(s, a)$ is the task depended reward and $r_c(s, a)$ is the penalty for violating the constraints. A description of each specific form of $r_t$ is given in the rest of this section.

### A.9.2 Air Hockey via Deep RL

The *RL3_polimi* team employs a DeepRL approach to train agents for the *defend* and *counter-attack* tasks. Both agents were trained employing the Soft Actor-Critic [SAC, Haarnoja et al., 2018] algorithm in combination with ATACOM [Liu et al., 2022]. These agents directly control the desired joint accelerations, which are then converted via ATACOM to joint position and velocities as shown in Figure 9. In this setting, they opted for the control of acceleration as it was difficult to train an agent that directly controls the joint positions while respecting the constraints, especially when it was transferred to the evaluation environment.

In order to comply with the constraints, the single integration mode of ATACOM proved to be insufficient as it consistently violated joint velocity limits. Despite encountering challenges in implementing the double integration setting of ATACOM, the team achieved success in ensuring that the agent adhered to the specified constraints. This was achieved by constraining joint accelerations within the range $[-1, 1]$. The combination of high-level control with the fixed conversion of the action space via ATACOM allowed for resolving the tasks while respecting the robot constraints.

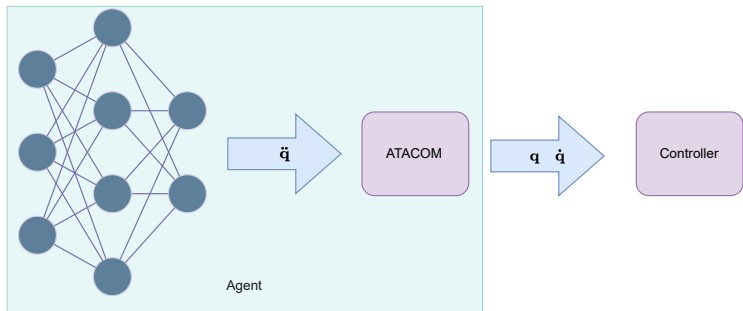

Figure 9: Architecture of the *RL3_polimi* Deep RL agents.

**Defend and Counter-attack**   *RL3_polimi* designs the *counter-attack task*, which is similar to the defend except for the possibility of scoring when blocking the puck. Indeed, for this task the agent is again penalized if a goal is scored against him and receives a bonus when touching the puck together with an additional bonus proportional to the puck velocity towards the opponent. Moreover, a high positive reward is received when after the puck interaction, a goal is scored. They trained two agents for the defend and counter-attack tasks. In these tasks, the episode starts with the puck moving towards the agent's goal. In both cases, the agent receives a high penalty (-100) when the puck breaches the goal. When the agent successfully interacts with the puck, a reward (+10) is granted, coupled with a reward proportional to the norm of the puck's velocity, with a negative sign when defending and positive sign when counter-attacking. This incentivizes the agent to interact with the puck and lower its velocity while also protecting the goal in the defend task and pushing back the puck in the counter-attack. Moreover, in the counter-attack, the agent can also score in the opponent's goal, for which he receives an additional large bonus of 1000.

**Return Home**   Moreover, *RL3_polimi* defines an additional task to train an agent to return to the initial position of the robot. At each step of the training process, the agent incurs a penalty based on the distance to the home configuration, $r = -\|\mathbf{q} - \mathbf{q}_{home}\|$. In order to combine these policies when building the high-level agent that plays the whole game, the initial states for this task are drawn from the distribution of terminal states of the other tasks

**Training via Curriculum Learning**   Curriculum Learning [CL, Narvekar et al., 2020] was applied to train each agent for the defend and counter-attack tasks. The return home task proved easy enough that no curriculum learning was needed to arrive at a good policy. Indeed, manual curriculums were devised across three levels of complexity: easy, medium, and hard tasks. These levels differ in the initial configuration of the puck, considering its position and velocity. In the easy task, the velocity along the y-axis was constrained and the velocity in the x-axis was kept low. Additionally, the puck was positioned at the end of the table, allowing the agent more reaction time. For the hard task, both x-axis and y-axis velocities were increased, and the puck's initial position was set in the middle of the table. The medium task served as an intermediary between the hard and easy tasks. For transitioning between different complexities a Sigmoid function was used to ensure the gradual increase of complexity.

**Training details**   The agents were trained for a total of 8000 epochs, where each epoch consists in 30 episodes of experience. During the first 3000 epochs the agent was trained with the easy task. After these initial epochs, a *soft* switch to the medium task was performed over the next 1000 epochs by increasing the proportion of the episodes drawn from the medium task. Between epoch 4000 to 6000, the agent was exclusively trained on the medium task. From epoch 6000 to 7000 the progressive

switch to the hard task was performed, and then the agent was trained exclusively on experience generated on the hard task.

After concluding the training in the defend task, the resulting policy was applied to the counter-attack one. Through curriculum learning, the task difficulty progressively increased to the hard level similar to the defend task. Employing this method, a counter-attack agent was successfully trained, capable of swiftly returning incoming pucks.

### A.9.3 RL-Optimized Rule-based Controllers

For the *hit* and *prepare* tasks, the *RL3_polimi* developed a Rule-Based Controller to be optimized via Policy Gradient Parameter-based Exploration [PGPE, Sehnke et al., 2008] following the same approach employed in [Likmeta et al., 2020]. In this case, the agent directly controls the position of the end-effector. The actions are transformed from the *task* space into the *joint* space, combining inverse kinematics and Anchored Quadratic Programming [AQP, Liu et al., 2021].

The rule-based policies were divided into 2 main branches that resemble the challenge tasks: *hit* and *prepare*. Each policy is further divided into phases, common to both of them:

- *Adjustment*: move the end-effector on (i) the line linking the puck and the goal for the *hit*; (ii) the vertical line parallel to the short side, passing through the puck, for the *prepare*;
- *Acceleration*: accelerate the end-effector and hit the puck. The *hit* policy hits the puck to score a goal, while the *prepare* one makes a more soft bump, in order to re-adjust the puck's position;
- *Final*: this phase only applies to the *hit* policy. Here, the agent keeps hitting the puck making it reach a desired acceleration, then slowly stops the end-effector following a curved trajectory.

In particular, the **hit policy** computes two quantities, $d\beta$ and $ds$ in the following way:

$$d\beta = \begin{cases} (\theta_0 + \theta_1 \cdot t_{phase} \cdot dt) \cdot correction \\ \frac{correction}{2} \\ (\theta_0 + \theta_1 \cdot dt) \cdot correction \end{cases}, ds = \begin{cases} \theta_2 & \textit{adjustment phase} \\ \frac{ds_{t-1} + \theta_3 \cdot t_{phase} \cdot dt}{radius + r_{mallet}} & \textit{acceleration phase} \\ constant & \textit{slow-down phase} \end{cases},$$

where $radius$ is the distance between the center of the puck and the end-effector, while $correction$ is always defined as:

$$correction = \begin{cases} 180 - \beta & y_{puck} \leq \frac{table\_width}{2} \\ \beta - 180 & y_{puck} > \frac{table\_width}{2} \end{cases}.$$

For what concerns the **prepare policy**, the quantities $d\beta$ and $ds$ are computed as:

$$d\beta = \begin{cases} \theta_0 \cdot t_{phase} + \theta_1 \cdot correction \\ correction \end{cases}, ds = \begin{cases} 5 \cdot 10^{-3} & \textit{adjustment phase} \\ \theta_2 & \textit{acceleration phase} \end{cases},$$

where $correction$ is defined as:

$$correction = \begin{cases} \beta - 90 & y_{puck} \leq 0 \\ 270 - \beta & y_{puck} > 0 \end{cases}.$$

All the quantities appearing in the computation of $d\beta$ and $ds$ are explained graphically in Figure 10 (left).

The major issue faced while developing this solution was the extreme sensitivity of the parameters w.r.t. the constraints. Indeed, the return landscape for our rule-based policies was highly non-smooth, i.e. a small variation could yield to a way bigger increase in the constraints violations. We employed PGPE to learn the parameters of the policies, but the learning process showed high sensibility to small parameters' variations.

In particular, for what concerns the *hit* task, the reward function is based on the construction of a triangle with the opponent's area and the puck as vertices (Figure 11). If the puck, after the hit, lies inside the triangle, a positive reward is assigned, proportional to the puck's velocity; if the puck is outside the triangle, a penalty is assigned. The more the puck is outside the polygon, the bigger the

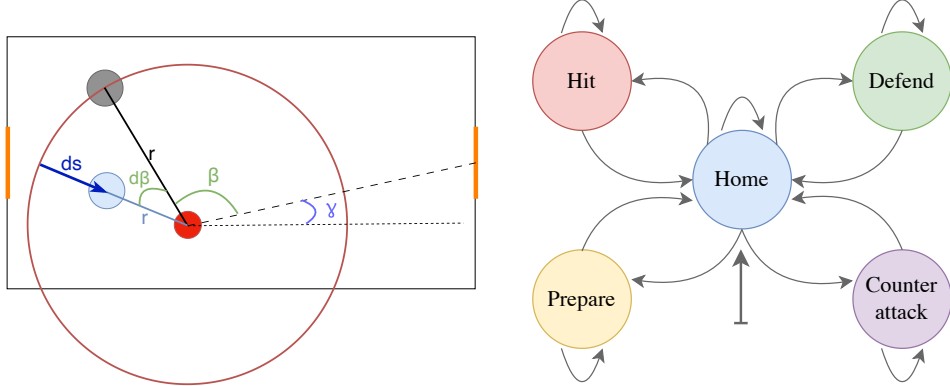

Figure 10: Components of the rule-based controllers: on the left, the Coordinates $d\beta$ and $ds$ used to find the next desired position of the end-effector. the puck is the red circle, the grey circle is the mallet and the blue circle is the next desired position of the mallet. On the right, the representation of the Finite State Machine for controlling task changes

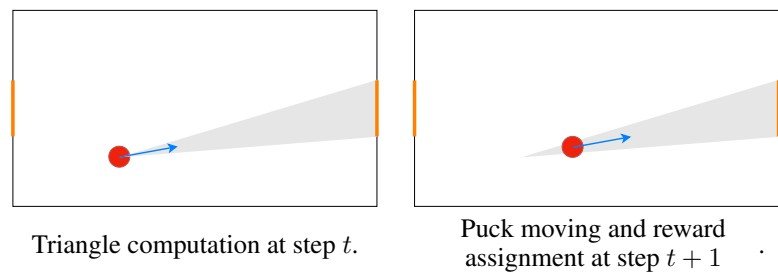

Triangle computation at step $t$.

Puck moving and reward
assignment at step $t+1$

Figure 11: Triangle construction for assigning reward after the mallet hit the puck.

penalty.

$$reward\_before\_hit = \begin{cases} -\frac{||ee_{pos}-puck_{pos}||}{0.5\cdot table\_diag} & \text{no hit} \\ A + B \cdot \frac{(1-(2\cdot\frac{\alpha}{\pi})^2)\cdot||v_{ee}||}{max\_vel} & \text{hit the puck} \\ \frac{1}{1-\gamma} & \text{goal} \end{cases}, \qquad (9)$$

Where $A$ and $B$ are constant values, respectively equal to 100 and 10, $table\_diag$ is the diagonal of the table, while $\alpha = \arctan 2(ee_{y\_vel}, ee_{x\_vel})$. Finally $max\_vel$ is a constant value equal to the maximum velocity observable in the environment.

If the agent hits the puck, it receives an instantaneous reward and, after that, the triangle approach is applied.

$$reward\_after\_hit = \begin{cases} B + ||v_{puck}|| & \text{puck inside the triangle} \\ -diff\_angle & \text{puck outside the triangle} \end{cases}, \qquad (10)$$

where B is a constant equal to 10 and

$$diff\_angle = \arctan 2(v_{y_{puck}}, v_{x_{puck}}) - angle\_border, \qquad (11)$$

where $angle\_border$ is the angle between the puck velocity vector and the closest triangle border.

### A.9.4   Noise Filtering via State Pre-Processing

In this section, we present how the *RL3_polimi* team treated the noise introduced in the evaluation environment. For training purposes, both the loss of the puck tracking and the noise in the observation were replicated in the local environment. To have a more realistic replica of the evaluation environment, an approximation of the noise model was performed. After the estimation, *RL3_polimi*

modified the local environment in order to make it return noisy observations so to locally evaluate their policies.

To smooth out the noise in the puck position and velocity observation, a simplified Kalman Filter, already provided within the challenge code, was employed.

In the Rule-based approach the de-noised observation is used to plan the next joint positions and velocities, by means of a combination of inverse kinematics and AQP solver. In order to delete the joint positions and velocities observation noise, the agent uses its output at the previous time step (instead of the observation). Such an approach works since the adopted techniques provide desired robot poses that are reachable. Indeed, on one side ATACOM makes the action space to contain only feasible actions, on the other side the same holds thanks to the AQP solver when translating the end-effector coordinates into the ones in the joint space. In the next step, these computed values are used in the agent's internal representation of joints state, overwriting the ones coming from the environment. This is possible since the proposed position is always made reachable thanks to the AQP solver. As one can imagine, the first observation is a noisy one and cannot be overwritten. However, the sensibility to such a noisy observation is negligible.

For what concerns the Deep RL agents, there is no need to overwrite the environment observation since, due to the ATACOM algorithm, the action space contains only feasible actions. No AQP solver is used in the Deep RL approach. As expected, the Rule-based policies were more sensitive to the noisy observations than the loss of tracking.

### A.9.5 Hierarchical agent

Finally, *RL3_polimi* combined the described solutions into a single agent which operates at a higher level, dynamically selecting the most appropriate task during the game. To select the task, this agent relied on two components: the *Switcher* and the *Finite State Machine* (FSM). The *Switcher* is the component developed to select a new task when the current one is completed. It consists in a simple rule-based controller which decides the task to employ via geometric considerations. After its last action, each task sends a signal to the high-level agent, notifying its completion. The switcher will select another task, which can potentially be also the same as before. The FSM is added as a second layer of safety to mitigate possible constraints violations while switching tasks. It forces the agent, at the end of each task, to go back to the home and start selecting a new task from there. The structure of the FSM can be observed in Figure 10. (right) The agent always starts a match in the *Home* state, it is possible to remain in the same state but not switching from a task to another without passing from the home state.

