# OpenReview forum: "A Retrospective on the Robot Air Hockey Challenge: Benchmarking Robust, Reliable, and Safe Learning Techniques for Real-world Robotics"
_NeurIPS.cc/2024/Datasets_and_Benchmarks_Track — NeurIPS 2024 Track Datasets and Benchmarks Poster_

### Official Review · Reviewer_qeQ2 · 2024-07-11
**Comprehensive Retrospective that Needs Some Reframing**

**Rating:** 6
**Confidence:** 3
**Correctness:** Documented in the full review.
**Clarity:** Documented in the full review.

**Review:**

Overall the paper is well written and provides a comprehensive overview, evaluation, and lessons learned stemming from the challenge itself. The paper does a good job at motivating the design of the challenge as well. The simulation environments are also released open source on the challenge website, although ease of accessing the links to the challenge benchmark tasks could be improved, which would enable others to test their algorithms in similar environments in the future. Finally, related work is well cited.

My major concern with the challenge in general is that the entire motivation hinges on real-world robotic testing and learning in real-world environments (e.g., to overcome sim2real gaps). However, the challenge itself was done entirely in simulation, and the resulting winning entries faced sim2real challenges. As such, it feels a bit disingenuous to motivate the paper in such a way. I was expecting the testing to be done in hardware through a remote mechanism at some stage of the challenge, similar to the TOTO effort (https://toto-benchmark.org/). I don't think this is a major flaw in the design of the competition, and in fact, I commend the authors in limiting team access to the ground truth environment etc. to mimic some of these issues. However, I feel that the structure of the paper and the way everything is motivated is currently misleading and that should be addressed.

**Strengths:**

Documented in the full review.

**Additional Feedback:**

Documented in the full review.

**Documentation:**

Documented in the full review.

**Ethics:**

No ethical concerns.

**Limitations:**

Documented in the full review.

**Opportunities For Improvement:**

Documented in the full review.

**Relation To Prior Work:**

Documented in the full review.

**Summary And Contributions:**

This paper provides a retrospective on the Robot Air Hockey Challenge which is itself effectively a benchmark task for ML applied to robotics applications. The paper both introduces the challenge, the benchmark environment in which the challenge took place (across a number of rounds), the results of the challenge, and lessons learned. The contributions are the design and release of the challenge, its benchmark environments, and the novelty of pushing for an ML applied to robotics benchmark that incorporated real hardware.

---

> ### Author Rebuttal · Authors · 2024-08-18
>
> We will try to deal with the reviewer’s concern in the final version of the paper by making our claims less strong. We agree that indeed a more in-depth evaluation of the real-world interaction would have been useful, but due to the complexity of the task (contact rich, fast motion, complex reset), this was not possible. However, as the reviewer stated, we tried to simulate the real-world issues these issues in
> the sim-to-sim setting. While this is definitively not enough for real-world deployment from scratch, we believe this challenge is an important step in that direction, going away from simple manipulation tasks. We will try to make this clear in the final version of the paper.
>
> We also thank the reviewer for bringing up the TOTO benchmark, which we will add to our citation list. However, we want to stress that the air hockey task requires a bit more effort to be fully automated for remote execution, as it requires fast dynamic movements and contacts. In future iterations of the challenge, we would definitively push in that direction. Note that in the challenge proposal, we were very clear about the real-world deployment, as we knew it would be extremely challenging.

---

### Official Review · Reviewer_zeVZ · 2024-07-23
**Great NeurIPS23 Challenge!**

**Rating:** 7
**Confidence:** 5
**Correctness:** Yes
**Clarity:** Yes, well written and easy to follow.

**Review:**

The challenge focuses on deploying machine learning solutions in real robotic environments, addressing practical challenges such as the sim-to-real gap, low-level control issues, safety problems, real-time requirements, and limited availability of real-world data. The challenge significantly impacts the field of robotics by encouraging the development of more robust and reliable learning techniques that can be deployed in real-world scenarios.

**Strengths:**

1. great challenge
2. a good review and summary of the challenge
3. well-written paper

**Additional Feedback:**

None

**Documentation:**

Yes.

**Ethics:**

No ethical concerns

**Limitations:**

Some more visual demo may be included

**Opportunities For Improvement:**

Some more visual demo may be included

**Relation To Prior Work:**

Yes

**Summary And Contributions:**

This paper is a review of NeurIPS 23 challenge: Robot Air Hockey Challenge. The challenge investigates and addresses the challenges of applying machine learning to real-world robotics. By using air hockey as a benchmark task, the competition required participants to tackle both low-level robotics problems and high-level tactical challenges.

---

> ### Author Rebuttal · Authors · 2024-08-12
>
> We thank the reviewer for his appreciation of our challenge and our retrospective.
> In our final version, we will try to address the other reviewers' concerns.

---

### Official Review · Reviewer_uZwi · 2024-07-24

**Rating:** 3
**Confidence:** 3
**Correctness:** Some claims are not appropriate or co…
**Clarity:** The paper written is ok, but with som…

**Review:**

The submission is of showcasing a well-structured and comprehensive approach to evaluating machine learning techniques in real-world robotics. The detailed organization of the Robot Air Hockey Challenge, including the division into stages and the use of both simulated and real-world environments.

Pros:
- Interesting Challenge: The Robot Air Hockey Challenge is an engaging and innovative real-world robotics competition that has successfully attracted participation from multiple teams.
- Well-Designed Rules and Metrics: The rules and evaluation metrics of the challenge are meticulously crafted and structured, ensuring a fair and comprehensive assessment of participants’ solutions.
- User-Friendly Simulator: The challenge provides a well-documented and easy-to-use simulator, making it accessible for participants to develop and test their algorithms.
- Real-World Robot Setup: The inclusion of a real-world robot setup allows for practical testing of participants’ algorithms, adding significant value to the competition.

Cons:
- Limited Real-World Evaluation: The primary limitation is the minimal real-world evaluation, which undermines the claims and contributions of the paper. The authors acknowledge that the real-world validation stage required extensive hand-tuning and engineering support, resulting in the competition outcomes being based solely on simulated tasks. This discrepancy makes the statement in the introduction that “the benchmark bridges the gap between real-world robotics and machine learning practitioners” seem exaggerated. With only three teams evaluated in the real-world setup and rankings based purely on simulation, the paper lacks sufficient evidence to demonstrate the improvements and distinctions of this work compared to other simulation benchmarks. It significantly deviates from the title “Benchmarking Robust, Reliable, and Safe Learning Techniques for Real-world Robotics” since the real-world evaluation is minimal and serves primarily as a demonstration.
- Simulation vs. Real-World Policy Efficacy: Another challenge is ensuring that the selected policies and methods in the simulation are indeed the best for real-world application. The competition’s three stages (qualifying, tournament, and real-world validation) predominantly evaluate policies in a simulated environment. However, due to the sim-to-real gap, the chosen policies might not be optimal for real-world scenarios due to imperfect simulations of dynamics and disturbances. The small number of teams (three) selected for real-world evaluation further limits the competition’s ability to identify the best real-world policies.

**Strengths:**

- An interesting challenge which has implemented as a real-world robot competition. There have been many teams participated in the challenge.
- The rules and evaluation metrics are well designed and structured.
- It has a well-documented and easy-to-use simulator for the challenge.
- It has a real-world robot setup which can test participants' algorithms using real robot.

**Additional Feedback:**

Please see the comments above.

**Documentation:**

Yes

**Limitations:**

- The key limitation is the limited real-world evaluation does not support the claims and the contributions of this paper. In section 2.1, Real-world validation stage the authors mention "Given that this stage required hand tuning and intense help and engineering from our side, the outcome of the competition relied only on the simulated tasks." This makes the statement in the introduction from the authors "the benchmark bridges the gap between real-world robotics and machine learning practitioners." to be exaggerated. Since there are only 3 teams having the chance to be evaluated in the real-world setup and the rankings of the challenge is purely based on simulated setup, it does not have enough evidence in the paper to show the improvements and differences of this work compared with other simulation benchmarks. It largely deviates from the title "Benchmarking Robust, Reliable, and Safe Learning Techniques for Real-world Robotics" since the real-world evaluation part is minimal and just for demo purposes.
- Another limitation of this challenge is how the selected policies and methods in simulation can be guaranteed as the best policies in the real-world competition. The current competition has three stages: 1) qualifying stage 2) tournament stage 3) real-world validation stage. The first two stages are evaluating the learned policies in the simulator. However, since there are sim-to-real gaps, there might be some possibilities for that the chosen policies would not be the best policies in the real-world due to imperfect simulations of dynamics and disturbance. The number of three selected teams seem a little bit small for the competition to find the best real-world policies for the challenge.

**Opportunities For Improvement:**

Major Improvements:
- There are some key limitations and flaws of this paper. Please check the limitation section.
- In abstract: the authors start with discussing the limited impact of machine learning methods in real-world robotics, where the topic is too big as a start point. Even with robot learning, it contains too many directions which cannot be explained in 1-2 sentences. For example, as the emergent conference of robot learning, CoRL, it has 1) Learning representations for robotic perception and control, 2) Learning robot foundation models or general-purpose knowledge systems for robotics, 3) Imitation learning for robotics, ... which has more than 15 topics of robot learning research. Authors simplify all these research as "robot learning" and call it "one of the most promising directions" would over-simplify the concepts and not specific enough for audience to understand what the paper is about to do. It would be great if authors can make the statements and scope of the paper to be more specific.
- In abstract: The authors state "The competition's results show that learning-based approaches with prior knowledge integration still outperform the data-driven approaches ...". It seems authors do not have clear understanding about the learning-based methods and data-driven methods. The comparison between the two is even not a fair comparison. First, representing methods as learning-based approaches and data-driven methods without any specificity do not make sense since there are a large overlap between these two types of methods. For example, imitation learning is a learning-based approach, but it also a data-driven approach since it requires data to train. The statement is not either rigorous nor accurate.


Minor Improvements:
- In the first paragraph of introduction, the authors state "purely data-driven approaches are still struggling in real-world robotics". There are two issues: 1) the examples showed by authors in the early context do not include any advancements of robotics using recent data-driven models, such as LLM or diffusion, 2) the authors do not show any evidence to support their claim on data-driven approaches are still struggling in real-world robotics. Authors should at least discuss about the existing models, such as RT from Google, and their issues. As well, at the end, the authors state "with the notable exception of quadruped locomotion", which seems pretty random and not rigorous since quadruped location is not the only exception of data-driven methods of robot learning which can adapt to the real-world.
- In the second paragraph of introduction, the first sentence is misleading, authors first state "robotics tasks are often the benchmark of choice in many areas of machine learning research" and then it criticize that "often the simulated setup is oversimplified". This is not a rigorous statement. Robot benchmarks include both simulation benchmark and real robot benchmark. It does not make sense to claim all robotics benchmarks have a big disconnection with the real-world based on only simulated setups.

**Relation To Prior Work:**

Yes

**Summary And Contributions:**

The paper presents a comprehensive overview of the Robot Air Hockey Challenge held at the NeurIPS 2023 conference. Below is a summary and key contributions of the submission:

Summary

The Robot Air Hockey Challenge was designed to evaluate and promote the development of robust, reliable, and safe machine learning techniques for real-world robotic applications. This challenge aimed to address several critical issues in robotics, such as the sim-to-real gap, low-level control problems, safety concerns, real-time requirements, and the limited availability of real-world data. By focusing on the dynamic and competitive task of robot air hockey, the challenge provided a platform for participants to demonstrate and benchmark their solutions against practical robotics problems.

Key Contributions

1. Benchmark Development:
   - The challenge introduced a new benchmark using the robot air hockey task, which requires participants to handle both low-level robotics problems and high-level tactics.
   - The benchmark is designed to address practical issues in robotics, including the sim-to-real gap, safety, real-time requirements, and limited real-world data.
2. Practical Implementation and Evaluation:
   - The competition involved a real-world robot air hockey setup with Kuka LBR IIWA 14 robots and an air hockey table, providing a realistic environment for testing the participants’ solutions.
   - A modified simulator incorporating real-world factors such as observation loss, tracking loss, model mismatch, and disturbances was used to evaluate the solutions.
3. Competition Structure:
   - The challenge was divided into three main stages: the Qualifying stage, the Tournament stage, and a non-ranking Real-world validation stage.
   - The Qualifying stage focused on individual sub-tasks (“Hit”, “Defend”, and “Prepare”), while the Tournament stage required participants to play full games of air hockey, integrating high-level skills with low-level control.
   - The Real-world validation stage tested the top solutions on the actual robotic platform, emphasizing the importance of safety and real-world applicability.
4. Evaluation Metrics:
   - Solutions were evaluated based on deployability and performance, with specific metrics for safety, end-effector position constraints, joint position limits, joint velocity limits, and computation time.
   - The competition results highlighted that learning-based approaches integrated with prior knowledge outperformed purely data-driven methods in building deployable robotics solutions.

---

> ### Author Rebuttal · Authors · 2024-08-18
>
> We thank the reviewer for the long and detailed review. We believe that the reviewer raised some important points. In the following, we will respond to all the reviewer’s concerns.
>
> > In abstract: the authors start with discussing the limited impact of machine learning methods in real-world robotics, where the topic is too big as a start point. Even with robot learning, it contains too many directions which cannot be explained in 1-2 sentences. For example, as the emergent conference of robot learning, CoRL, it has 1) Learning representations for robotic perception and control, 2) Learning robot foundation models or general-purpose knowledge systems for robotics, 3) Imitation learning for robotics, ... which
> has more than 15 topics of robot learning research. Authors simplify all these research as ”robot learning” and call it ”one of the most promising directions” would over-simplify the concepts and not specific enough for audience to understand what the paper is about to do. It would be great if authors can make the statements and scope of the paper to be more specific.
>
> We agree with the reviewer that robot learning is a very wide term. However, in the context of the abstract, we want to highlight that applying learning techniques to robotics is a key factor in improving the state-of-the-art. In fact, our challenge has a very broad scope, and it is not limited to any specific robot learning technique. In fact, the submissions to this year’s challenge already covered topics like
> representation learning, imitation learning, model-based/free reinforcement learning, safe reinforcement learning, etc. Moreover, in future challenge iterations, we expect to see approaches that integrate, e.g., large language models and perception.
>
> > In abstract: The authors state ”The competition’s results show that learning-based approaches with prior knowledge integration still outperform the data-driven approaches ...”. It seems authors do not have clear understanding about the learning-based methods
> and data-driven methods. The comparison between the two is even not a fair comparison. First, representing methods as learning-based approaches and data-driven methods without any specificity do not make sense since there are a large overlap between these two types of methods. For example, imitation learning is a learning-based approach, but it also a data-driven approach since it requires data to train. The statement is not either rigorous nor accurate.
>
> We believe that the reviewer misunderstood this sentence. Here we are comparing methods that are purely data-driven and methods that also exploit some inductive biases (i.e., “prior knowledge“), mostly coming from robotics, such as kinematics, dynamics, and geometry. We will try to clarify this further in the final version of the paper.
>
> > In the first paragraph of introduction, the authors state ”purely data-driven approaches are still struggling in real-world robotics”. There are two issues: 1) the examples showed by authors in the early context do not include any advancements of robotics using recent
> data-driven models, such as LLM or diffusion, 2) the authors do not show any evidence to support their claim on data-driven approaches are still struggling in real-world robotics. Authors should at least discuss about the existing models, such as RT from Google, and their issues. As well, at the end, the authors state ”with the notable exception of quadruped locomotion”, which seems pretty random and not rigorous since quadruped location is not the only exception of data-driven methods of robot learning which can adapt to the realworld.
>
> We thank the reviewer for pointing this out. We will include a discussion on the limitations of this LLM model. In our opinion, it is clear that this model cannot yet handle highly dynamic and contact rich tasks (even only considering computational reasons). Also, it is important to highlight that even this model relies heavily on some predefined behaviors exploiting robotics knowledge to allow proper deployment. Locomotion, instead, is an area where pure data-driven approaches outperformed and fully replaced existing robotics baselines. However, we agree with the reviewer that the claim is too strong, and we will tone it down.

---

> > ### Author Response · Authors · 2024-08-18
> > **Responses (2/3)**
> >
> > > In the second paragraph of introduction, the first sentence is misleading, authors first state ”robotics tasks are often the benchmark of choice in many areas of machine learning research” and then it criticize that ”often the simulated setup is oversimplified”. This is not a rigorous statement. Robot benchmarks include both simulation benchmark and real robot benchmark. It does not make sense to claim all robotics benchmarks have a big disconnection with the real-world based on only simulated setups.
> >
> > We want to highlight that from a robotics point of view, we are indeed not aware of a benchmark that thoroughly encompasses all the issues that we highlight in this challenge, such as safety, dynamic environment, fast and reactive controllers with limited computation budget, and limited target environment data. However, in light of the reviewer’s comment we agree that we should make our statements more precise. In fact, simulated robotic benchmarks are appealing as they do not necessitate for any investments in real robotic hardware. Yet, these purely simulated challenges typically do not consider challenges that are crucial for sim-to-real transfer, such as computational requirements, safety constraints, etc. While some robotic challenges also address the real system directly or provide both simulation and real-world evaluation (e.g., the real robot challenge, ManiSkill challenge). The tasks are quasi-static and pratical issues such as safety are not addressed. We will therefore refine the statements in the final version of the paper as it was not our intention to claim that all simulated robotics benchmarks have a big disconnection with the real world, however, things like dealing with a highly dynamic task, limited compute budget and real-time execution are typically not reflected in such benchmarks, making real-world deployment challenging.
> >
> > > The key limitation is the limited real-world evaluation does not support the claims and the contributions of this paper. In section 2.1, Real-world validation stage the authors mention ”Given that this stage required hand tuning and intense help and engineering from our side, the outcome of the competition relied only on the simulated tasks.” This makes the statement in the introduction from the authors ”the benchmark bridges the gap between real-world robotics and machine learning practitioners.” to be exaggerated. Since there are only 3 teams having the chance to be evaluated in the real-world setup and the rankings of the challenge is purely based on simulated setup, it does not have enough evidence in the paper to show the improvements and differences of this work compared with other simulation benchmarks. It largely deviates from the title ”Benchmarking Robust, Reliable, and Safe Learning Techniques for Real-world Robotics” since the real-world evaluation part is minimal and just for demo purposes.
> >
> > We disagree with the reviewer on this point. We want to highlight that the key issue for deploying participants’ solutions on such a complex platform is the safety aspect. In fact, our thorough benchmarking in simulation revealed that only three teams were able to successfully fulfill the safety requirements for all matches. We believe that one core message from our challenge is that safety aspects should be better reflected when designing learning-based approaches for real-world robotics. We will further emphasize this aspect in the final version of the paper. Moreover, we consider it a success of our simulation phase that we could successfully run the three approaches on the real system with only minor additional engineering (e.g. detecting a flipped mallet). While it would have been possible to deploy more solutions, this would have requested an unreasonable amount of engineering, making the deployed solution too far from the one in the simulation. In future iterations of the challenge, we, therefore, want to improve the focus on the safety aspects such that more solutions can actually be deployed on the real robot.

---

> > > ### Author Response · Authors · 2024-08-18
> > > **Responses (3/3)**
> > >
> > > > Another limitation of this challenge is how the selected policies and methods in simulation can be guaranteed as the best policies in the real-world competition. However, since there are sim-to-real gaps, there might be some possibilities for that the chosen policies would not be the best policies in the real-world due to imperfect simulations of dynamics and disturbance.  The number of three selected teams seem a little bit small for the competition to find the best real-world policies for the challenge.
> > >
> > > While the reviewer raises an important point, this is an issue we considered in the challenge. Indeed, our setting assumes sim-to-sim gap between simulation and deployment environments, and the participants were forced to produce algorithms robust enough for the target environment with limited data coming from it. If we had planned a real-world stage, we would have followed exactly the same procedure to deal with sim-to-real gap. It is also worth noting that the results in the simulation closely reflected the results in the real robot setting. Indeed the team that won the challenge proved to be the best in terms of deployability. The third classified team performed better than the second one. However, it must be considered that the third team had a bug during the first round of the evaluation, causing faulty resets and constraint violations, which caused the team to lose many games.
> > >
> > > Finally, we want to remark that in the competition proposal, we made it clear we would only evaluate the three most successful approaches. This is because real-world deployment is incredibly challenging and it is still an open question of how to deal with this issue properly. We believe that this challenge is a good step forward, and the real-world deployment of the participant’s solutions went beyond our expectations.

---

### Official Review · Reviewer_neoz · 2024-07-25

**Rating:** 9
**Confidence:** 4
**Correctness:** Yes, looks comprehensive.

**Review:**

Strengths

* **Real-world Focus.** The challenge goes beyond simulation to address real-world robotics tasks, highlighting practical issues that are often neglected in purely academic settings.

* **Integration of Classical and Modern Techniques.** The use of classical robotics techniques, such as Model Predictive Control (MPC), alongside modern learning-based methods, demonstrates a well-rounded approach that leverages the strengths of both domains.
Comprehensive Benchmarking: The challenge covers a wide range of tasks from low-level control to high-level tactics, providing a thorough evaluation of the participating solutions.

* **Platform for Fair Competition.** The challenge offers a fair and competitive platform for determining state-of-the-art approaches in robot control and learning, fostering innovation and collaboration within the community.

Weaknesses:
* **Limited Real-world Data Collection.** The hardware availability constraints limited the amount of real-world data that could be collected, which is crucial for improving sim-to-real transfer.

* **External Motion Tracking.** The use of external motion tracking systems overlooks the challenges of onboard perception and localization, which are critical for real-world applications.

* **Computationally Expensive Algorithms.** Many of the learning algorithms used are computationally intensive. Although the authors acknowledge the delays in processing and their impact on safety, they should have included metrics that capture hardware efficiency along with performance metrics.

* **Sample Size and Statistical Analysis** The conclusions are based on a limited sample size, which might affect the generalizability of the findings. The statistical analysis could be more robust to enhance the reliability of the results.

* **Safety and Robustness.** The penalty-based safety specifications were found to be brittle in out-of-distribution situations, indicating a need for more robust safety mechanisms.

**Strengths:**

Please see the strengths in the review above.

**Additional Feedback:**

Please see my comments in the review and areas of improvement.

**Clarity:**

Yes, it is easy to read. The appendix has some interesting information about different aspects of the challenge.

**Documentation:**

Yes. The website has enough information and pointer to readthedocs. I like the leaderboard. It is important to have leaderboards on standard problems to benchmark and determine state of the art techniques. I just hope the authors continue to invest and expand on these challenges.

**Ethics:**

No ethical issues found.

**Limitations:**

Please see my comments in the weakness section of the review.

**Opportunities For Improvement:**

Please see my comments in the weakness. The authors should incorporate metrics that capture hardware efficiency, computational costs, and processing delays. Including these metrics will provide a more comprehensive evaluation of the solutions and highlight the practical challenges in deploying these systems in real-world applications

**Relation To Prior Work:**

Yes, looks adequate.

**Summary And Contributions:**

The authors presents a detailed retrospective analysis of the Robot Air Hockey Challenge held at NeurIPS 2023. It focuses on benchmarking learning techniques for real-world robotics, emphasizing the integration of classical robotics methods with modern learning-based approaches. The challenge addressed key issues such as the sim-to-real gap, safety, real-time requirements, and limited data availability. The findings suggest that combining prior knowledge with learning-based methods outperforms purely data-driven approaches, providing valuable insights for future research and competitions in robotic learning.

---

> ### Author Rebuttal · Authors · 2024-08-18
>
> We thank the reviewer for the detailed review and the appreciation for our work. In the following, we handle the five weaknesses highlighted by the reviewer.
>
> **Limited Real-world Data Collection**
>
> The problem of limited real-world data is unfortunately a structural issue of the challenge. Indeed, if the task is complex enough and the experimental setup is quite complex, like the one we are using, it is difficult to have a massive amount of real-world data. While we agree with the reviewer that this is a key point to improve in future challenges, the limited availability of real-world data is an inherent challenge that needs to be faced in any real-world robotics experiment/setup.
>
> **External Motion Tracking**
>
> The reviewer has highlighted another key point.  As stated in the limitation, we decided to remove this aspect from the challenge as playing a full air hockey game was already difficult enough. We believe that thanks to the new baselines of the participants, it will be easier to include this important factor in future iterations.
>
> **Computationally Expensive Algorithms**
>
> We want to clarify that the execution time and the capability of keeping real-time execution were directly measured in the challenge and considered in the penalties. In other words, failing to meet the computational requirements reduced the team’s scores. We will try to make this clearer in the final version of the paper. Notice that the most computationally expensive approaches relied on other techniques to allow real-time deployment, such as imitation learning.
>
> **Sample Size and Statistical Analysis**
>
> We fully agree with the reviewer, and we hope to have more participants in the next challenge to allow us to draw a stronger conclusion on the different methodologies. Unfortunately, some of the teams dropped their participation due to strict safety requirements. We will also update the evaluation metric to encourage more active participation in the future. Another possible direction is to increase the number of games played against each opponent in the simulated setting.
>
> **Safety and Robustness**
>
> While we agree with the reviewer that having better safety measures is important, in this part of the paper, we refer to specific safety methodologies that were unable to handle out-of-distribution safety. We will try to clarify the sentence in the final version of the paper. However, we will for sure improve the safety measurement in the next challenge iterations

---

> > ### Comment · Reviewer_neoz · 2024-08-30
> > **Response to the authors**
> >
> > Thanks for directly addressing the limitations in my review. While this paper is the culmination of the air hockey challenge in 2023, I also encourage the authors to continue this competition with specific emphasis on each challenge. I will retain my existing score.

---

### Official Review · Reviewer_3fJX · 2024-08-04
**Interesting paper but is very limited in analysis**

**Rating:** 6
**Confidence:** 4
**Correctness:** yes
**Clarity:** yes

**Review:**

Strength:
-  I really like the motivation of introducing this competition, where safety, imperfect models, limited data, reactiveness, and observation noises and disturbances are indeed critical challenges that blocks the robot learning based approaches being applied in the real world.
- It is nice to see the competition has received wide participations and the authors have reported the detailed metrics of all existing team on the competition, which shed light on the existing results on the competition
- The detailed framework of the highest score participant - AiRLIHockey is carefully explained, which provides a good baseline for subsequent participations.
- The takeaways for the future research directions are important to share with the community

Weakness:
- The mathematical problem formulation is poorly explained, including the state space, action space, detailed metric design and computation for the qualifying stage, and etc.
- The pros and cons of the participating team methodology is not delivered in detail. As retrospective paper, I would like to see the detailed analysis of each team core innovation and the reason why they couldn’t reach a better score. Now existing team solutions are mixed together, it is hard to grasp important takeaways.

**Strengths:**

see the Review section

**Additional Feedback:**

Overall good quality, more insights and mathematically backgrounds need to be established.

**Documentation:**

yes

**Ethics:**

no concern

**Limitations:**

yes

**Opportunities For Improvement:**

- make the mathematical formulation of this challenge more rigorous and explicit
- explicitly summarize the evidence of key takeaways and pros / cons for each participant techniques.

**Relation To Prior Work:**

yes

**Summary And Contributions:**

Machine learning has significantly impacted many fields, but its application in robotics is still limited due to real-world challenges. To address these, the Robot Air Hockey Challenge was organized at NeurIPS 2023, using air hockey as a benchmark to cover both low-level and high-level robotics problems. Participants faced practical issues like the sim-to-real gap, control, safety, real-time requirements, and limited data. Results showed that approaches integrating prior knowledge outperformed purely data-driven methods. The competition provided insights into overlooked real-world factors and established a foundation for future research and competitions.

---

> ### Author Rebuttal · Authors · 2024-08-18
>
> **Problem Formulation**
>
> We thank the reviewer for the important remarks. In the final version of the paper, we will make sure to add a section describing the state space and the metrics in detail. In the initial submission, we left them out as they were discussed in detail in the documentation, but we agree that these details are really important, and we will make sure to discuss them in detail.
>
> **Pros and Cons for the participation teams**
>
> We agree that the discussion of pros and cons is a bit condensed due to the page limit. We will use the extra page for the final submission to expand this section and make sure the main takeaways are discussed clearly enough.

---

### Official Review · Reviewer_w1oz · 2024-08-07
**A great retrospective rather than a dataset or a benchmark paper**

**Rating:** 5
**Confidence:** 3
**Correctness:** Yes.
**Clarity:** This paper is well-written and provid…

**Review:**

This is certainly a good retrospective paper; my only concern is whether it is a dataset or benchmark paper.
While it introduces an air-hockey environment, I'm slightly concerned that it might be too narrow for benchmarking all these techniques for real-world robotics as the authors claim.

**Strengths:**

- The paper introduces a benchmark platform by using the Robot Air Hockey Challenge, which encompasses both low-level control and high-level tactical issues and addresses a critical obstacle in deploying machine learning solutions in real-world robotics.
- The paper discusses the relevance of addressing practical challenges like safety, real-time requirements, and limited data in robotics to both academic research and industrial robotics.
- The paper also provides a thorough analysis of the competition results, including ablation studies and detailed evaluations of various solutions.

**Additional Feedback:**

It would be helpful if the authors could submit their paper using the template with line numbers, allowing reviewers to easily reference specific locations in the paper.

**Documentation:**

Yes.

**Ethics:**

No other ethical concerns.

**Limitations:**

The authors have addressed the limitations and potential negative societal impacts.

**Opportunities For Improvement:**

It would be beneficial to have more environments for benchmarking techniques in real-world robotics, which could makes the paper more convincing.

**Relation To Prior Work:**

Yes.

**Summary And Contributions:**

The paper reviews the Robot Air Hockey Challenge at NeurIPS 2023, which aimed to bridge the gap between machine learning and real-world robotics by using an air hockey task as a benchmark. Participants addressed various practical challenges, such as the sim-to-real gap, low-level control issues, safety problems, real-time requirements, and limited data availability. The results indicated that learning-based approaches integrating prior knowledge outperformed purely data-driven methods. The paper discusses the structure and outcomes of the competition, key insights gained, and the performance of top teams.

---

> ### Author Rebuttal · Authors · 2024-08-12
>
> We thank the reviewer for appreciating our retrospective. We agree that this is not a classical dataset and benchmark paper. Indeed, as the reviewer said, we would require a higher benchmark for this to be a contribution to the dataset and benchmark track.
>
> However, we want to point out that this is a "NeurIPS23 Competition Analysis Paper". Indeed, for the 2023 competitions, there was a (new) requirement to submit to the Dataset and benchmark track:
>
> https://neurips.cc/Conferences/2023/CallForCompetitions
>
> > PROCEEDINGS (NEW!)
> >
> > This year, there are new publication requirements for competition reports co-authored by both organizers and participants. Unlike previous years, accepted competitions in 2023 will be required to submit their post-competition analyses as papers to the 2024 NeurIPS D&B track (next year). In order to ensure that the competition proposal and results are consistent, reviewers in 2024 will have access to the 2023 proposals. To minimize experimental bias, any deviations from the proposal will need to be justified.
>
> Therefore, adding more benchmarks would be out of scope, given that this is a competition analysis. If the reviewer has further comments on the competition analysis, we will be happy to respond to any doubt or improve the analysis following the reviewer's suggestions.

---

### Decision · Program_Chairs · 2024-09-26

**Decision:**

Accept (Poster)

**Comment:**

This paper presents a retrospective on the Robot Air Hockey Challenge at NeurIPS 2023. This retrospective analyzes the performances of classic methods and learning based methods, and provides insightful findings.

As pointed out by reviewers, this paper has some problems of the inaccurate presentation and over claim, which should be addressed in the final version.